# Measuring And Improving Engagement Of Text-to-Image Generation Models

**Varun Khurana**[*] **Yaman K Singla**[*] **Jayakumar Subramanian**

**Changyou Chen** **Rajiv Ratn Shah** **Zhiqiang Xu** **Balaji Krishnamurthy**

Adobe Media and Data Science Research, SUNY at Buffalo, IIIT-Delhi, MBZUAI

## Abstract

Recent advances in text-to-image generation have achieved impressive aesthetic quality, making these models usable for both personal and commercial purposes. However, in the fields of marketing and advertising, images are often created to be more engaging, as reflected in user behaviors such as increasing clicks, likes, and purchases, in addition to being aesthetically pleasing. To this end, we introduce the challenge of optimizing the image generation process for improved viewer engagement. In order to study image engagement and utility in real-world marketing scenarios, we collect *EngagingImageNet*, the first large-scale dataset of images, along with associated user engagement metrics. Further, we find that existing image evaluation metrics like aesthetics, CLIPScore, PickScore, ImageReward, *etc.* are unable to capture viewer engagement. To address the lack of reliable metrics for assessing image utility, we use the *EngagingImageNet* dataset to train *EngageNet*, an engagement-aware Vision Language Model (VLM) that predicts viewer engagement of images by leveraging contextual information about the tweet content, enterprise details, and posting time. We then explore methods to enhance the engagement of text-to-image models, making initial strides in this direction. These include conditioning image generation on improved prompts, supervised fine-tuning of stable diffusion on high-performing images, and reinforcement learning to align stable diffusion with *EngageNet*-based reward signals, all of which lead to the generation of images with higher viewer engagement. Finally, we propose the *Engagement Arena*, to benchmark text-to-image models based on their ability to generate engaging images, using *EngageNet* as the evaluator, thereby encouraging the research community to measure further advances in the engagement of text-to-image modeling. These contributions provide a new pathway for advancing utility-driven image generation, with significant implications for the commercial application of image generation. We have released our code and dataset on `behavior-in-the-wild.github.io/image-engagement`.

## 1 Introduction

Machine learning models that interact with humans are built as a means to achieve an end, and performance metrics in their respective fields reflect how effectively these models meet the ends. For instance, recommendation systems are optimized to capture maximum viewer interest and the key performance metrics tracked by the research community are clickthrough rates and the number and ranking of relevant documents recommended out of the total document set (Bobadilla et al., 2013). Similarly, chat assistants are optimized for being helpful, and the commonly tracked metrics are the scores of responses preferred by humans (Ouyang et al., 2022; Stiennon et al., 2020). In the case of image generation, industries such as e-commerce, fashion, education, and advertising aim to optimize user-focused outcomes like clicks, purchases, retention, and user engagement. However, the metrics used by the image generation research community often emphasize *aesthetic appeal* (Xu et al., 2024; Kirstain et al., 2023; Black et al., 2023) and *realism* (Dhariwal & Nichol, 2021; Saharia et al., 2022; Ho et al., 2020; Rombach et al., 2022) factors that crucial for image acceptability but not necessarily aligned with the ultimate goals of viewer engagement.

---

[*]Equal Contribution. Contact behavior-in-the-wild@googlegroups.com for questions and suggestions.

Figure 1: Some images from the EngagingImageNet dataset. We constructed pairs of similar images posted within a 45 days interval by the same account. In each pair shown in the figure, the left image corresponds to lower likes and the right one received higher likes. However, existing image generation metrics like Aesthetics, PickScore, Human Preference Score, ImageReward, *etc.*, exhibit image preference in the opposite direction as actual user engagement.

We find that popular image generation metrics such as Aesthetics (Schuhmann et al., 2022), ImageReward (Xu et al., 2024), Human Preference Score (HPS) (Wu et al., 2023), and CLIP-H (Radford et al., 2021) have a correlation ranging from 0.02-0.08 with user engagement measured by likes, roughly equal to random chance (Table 2). Fig. 1 illustrates this effect through some randomly picked high and low engagement image samples. Further, one may think that the preferences of image creators (*e.g.*, in the form of upvotes on platforms like Pick-a-Pic (Kirstain et al., 2023) or Discord (Wu et al., 2023)) are a good estimate of image-consumer engagement. However, we find that PickScore and HPS, the reward models trained on a large dataset of creator preferences, correlate 0.07 with user engagement. Therefore, there is a lack of reliable metrics capturing viewer engagement on images.

The lack of progress can largely be attributed to the absence of a large and open dataset of customer engagement metrics over images. The most common image generation datasets, MS-COCO (Lin et al., 2014) and LAION (Schuhmann et al., 2022), contain no signals for user engagement. Therefore, to spur research in the direction of measurement and optimization of image generation for user engagement, we curate a large-scale dataset, **EngagingImageNet**. EngagingImageNet (§2) consists of 168 million tweets capturing 17 years of high-quality enterprise images for over ten thousand brand accounts and average user engagement of images in the form of likes*. We release EngagingImageNet to serve as a starting point for measuring, benchmarking, and modeling large-scale engagement-optimized image generation.

**EngageNet as a scoring function to score engagement:** EnagingImageNet allows us to train a scoring function that estimates the user engagement on a particular generated image. We formulate this problem as simulating the engagement in the form of user likes over an image-containing tweet (§3.2). We carry out visual instruction finetuning of LLaVA-1.5 13B (Liu et al., 2023) model to estimate the brand-normalized likes given the image along with contextual information that includes input account handle, image description, and time of the tweet. We find that the resulting scoring model, **EngageNet**, achieves a high correlation of 0.62 with actual user engagement.

**Engagement Arena:** Next, leveraging EngageNet as a judge, on the lines of LMSYS arena (Zheng et al., 2024; Chiang et al., 2024), we propose **Engagement Arena**, an arena where we test the engagement of images generated by various image generation models for the same prompt. Using EngageNet's reward estimates, we compute Elo ratings of a number of popular open-source text-to-image generation models, including Stable Diffusion-3 (Esser et al., 2024), Flux.1-dev (Labs, 2024), Stable Diffusion-XL (Podell et al., 2023), SDXL-DPO (Wallace et al., 2024), PixArt-alpha (Chen et al., 2024b), Pixart-sigma (Chen et al., 2024a), Stable Diffusion 2.1 (Rombach et al., 2022), *etc*, and closed-source models like DALL.E-2 (Ramesh et al., 2022). Further, we encourage the research community to adopt Engagement Arena as a basis for measuring further advances in the engagement capabilities of text-to-image modeling and incorporating user engagement into the learning process.

**Optimizing the Image generation process with the goal of increasing engagement:** Finally, we explore train-time and run-time methods to induce the goal of engagement in the text-to-image generation process: (1) Run-time: conditioning the diffusion model on prompts aligned with higher user engagement, (2) Train-time: fine-tuning the diffusion model on high-engagement images, and (3) Train-time: aligning the diffusion model with EngageNet-based rewards via reinforcement learning. We present the results of these experiments in Section 4 and report the efficacy of each method in generating more engaging images.

To summarize, we make the following contributions:

---

*EngagingImageNet was collected using Twitter API over a period of several years.

Figure 2: Figure illustrating the steps involved in the creation of the EngagingImageNet dataset.

1. We introduce the problem of engagement-optimized image generation. Images, especially in industries like advertising, fashion, and e-commerce, are created to achieve user engagement in the form of clicks, likes, and purchases. Therefore, the image generation process needs to be biased on the image's eventual utility, in addition to the common goals of high aesthetics and fidelity.

2. We curate **EngagingImageNet**, a large-scale, high-quality dataset consisting of user engagement over images. EngagingImageNet consists of 168 million tweets collected from 10,135 enterprise Twitter accounts from the time period 2007 to 2023. It consists of the account name, tweet text, media posted with the tweet, image captions, keywords, colors and tones, the time of posting, and the number of likes the image received. The dataset is instrumental in our study of image engagement as the utility in real-world marketing scenarios.

3. We train an engagement-aware vision language model (VLM), called **EngageNet**, to predict user engagement over images. EngageNet exhibits strong performance in estimating user engagement compared to other commonly used metrics like FID and aesthetics for evaluating the performance of text-to-image generation models as well as state-of-the-art LLMs like GPT-3.5 and GPT-4V.

4. Using EngageNet's predicted engagement scores as a reward, we introduce **Engagement Arena**, the first automated arena to benchmark the engagement of text-to-image models. We rank several popular text-to-image models on their ability to generate engaging images and further encourage the community to submit their models to the arena.

5. We demonstrate introducing the goal of engagement in the text-to-image generation process. We present several approaches to achieve this. These include conditioning of text-to-image generation on prompts corresponding to high user engagement, supervised fine-tuning of stable diffusion on high-engagement images, and reinforcement learning to align stable diffusion with EngageNet-based rewards, all of which lead to the generation of more engaging images to varying degrees.

## 2  *EngagingImageNet*: DATASET WITH IN-THE-WILD USER ENGAGEMENT

To gain insights into image engagement and align text-to-image generation with user engagement, we start by collecting a large dataset of user engagement over images. Our data collection method involved leveraging Twitter, a platform extensively utilized by brands for various purposes such as ongoing product campaigns, sales, offers, discounts, brand building, and community engagement (Alalwan et al., 2017). Twitter user engagement metrics encompass user likes, retweets, comments, mentions, follows, clicks on embedded media, and links. However, the Twitter API provides access to only user likes, retweets, and comments for a given post, with access to comments necessitating a separate and costly call. Therefore, utilizing API licenses, we extracted the following data from Twitter: Tweet ID, company name, username, timestamp, tweet text, media files, and user likes.

We focus on enterprise handles for our data collection efforts since the content released by enterprises has the explicit goal of user engagement and is relatively much cleaner than user-generated content. We began by compiling a comprehensive list of company names using the Wikidata knowledge graph (Wikidata contributors, ongoing), focusing on entities categorized as 'business' or 'enterprise'. We conducted Google searches to gather a list of all associated accounts for these companies. For example, for Adobe, this encompassed accounts like Adobe, Adobe Photoshop, Adobe Lightroom, Adobe Experience Cloud, and so forth. This method enabled us to amass a total of 10,135 enterprise Twitter handles. We then utilized the Twitter API to retrieve tweets posted by these enterprises spanning from 2007 to the closure of the Twitter API in January 2023. This effort resulted in the collection of 168 million tweets over a 17-year period, with 28.5 million of these tweets featuring various forms of media, including GIFs, images, and videos. Fig. 9 shows several examples of media and tweets present in the EngagingImageNet.

Next, for each username, we bin the tweets falling in the bottom 60 percentile (and having absolute likes $> 20$), 60-90 percentile (and having absolute likes $> 30$), and 90-100 percentile (and having

Table 1: A comparison of datasets containing image preferences

| Dataset | Size |
|---|---|
| Pick-a-Pic (Kirstain et al., 2023) | 968,965 rankings originated from 66,798 prompts and 6,394 users |
| Human Preference Score (Wu et al., 2023) | Total of 98,807 images generated from 25,205 text prompts |
| ImageReward (Xu et al., 2024) | Annotations for 8878 text prompts and corresponding model outputs sampled from DiffusionDB, resulting in 136,892 compared pairs |
| EngagingImageNet (Ours) | **28.5 million** tweets containing media, captions, colors, tones, and objects, and with user likes as engagement metric |

absolute likes $> 40$) of all tweets per account based on the number of user likes. These buckets are subsequently referred to as 'low', 'medium' and 'high' liked buckets, respectively. The resulting dataset, EngagingImageNet, consists of 837,532 samples, having 144,905 high-liked images, 336,200 medium-liked images and 356,427 low-liked images. The high-liked tweets had an average of 2435 likes, while low-liked tweets had an average of 132 likes. Subsequently, all images were verbalized by extracting their captions using LLaVA (Liu et al., 2023), colors and tones along with their coverage using (Qin et al., 2020). Further details regarding the data processing are provided in Appendix E and G.

**Ethics Statement**: We have taken measures to ensure that the EngagingImageNet dataset does not contain any personally identifiable information (PII). The dataset contains only public, non-individualized content following ethical considerations, data protection guidelines, and Twitter API guidelines. To ensure these, we: (1) Collect tweets and aggregate user likes from accounts identified using the Wikidata Knowledge Graph and marked as "enterprise" or "business", (2) Don't collect any personal or sensitive user information, (3) Remove references to specific users or personal identifiers (such as through using account handles), (3) Collect only aggregate engagement metrics (e.g., overall user likes data) to analyze engagement trends without tracking or identifying individuals. As part of our commitment to responsible research, EngagingImageNet will be open-sourced in phases, starting with smaller controlled releases, while actively engaging with the research community to identify and address any potential concerns. We will continue to review compliance with data protection guidelines and encourage the adoption of similar ethical best practices within the community. We discuss this in more detail in Appendix L.

## 3 *EngageNet*: MEASURING IMAGE ENGAGEMENT

Simulating engagement is inherently challenging, as prior research suggests that both experts and non-experts struggle to accurately predict engagement outcomes (Tetlock, 2017; for, 2023; Tan et al., 2014; Isola et al., 2013; Singh et al., 2024). These findings highlight the necessity of an automated and reliable metric for estimating image engagement. In this section, we cover the alignment of existing metrics with viewer engagement and design a model to measure the progress of image generation models with the goal of engagement.

### 3.1 ALIGNMENT OF EXISTING MODELS WITH VIEWER ENGAGEMENT

In order to measure the engagement potential of generated images, we first check the alignment of the most popular existing metrics used for evaluating text-to-image models with viewer engagement. For this, we calculate the Pearson correlation of Aesthetic score (Schuhmann et al., 2022), CLIP score (Radford et al., 2021), Pickscore (Kirstain et al., 2023), ImageReward (Xu et al., 2024), and Human Preference Score (HPS) (Wu et al., 2023) with ground truth brand-wise normalized user likes (0-100) from EngagingImageNet. Table 2 presents the results of this analysis. Clearly, the existing metrics are not aligned with user engagement.

This non-alignment can be attributed to the following reasons. Models like PickScore, HPS, and ImageReward are optimized for creator preferences captured on platforms like Discord or custom web applications rather than user (viewer) preferences. The feedback from image creators or communities on these platforms tends to reflect artistic or stylistic biases that do not necessarily correlate with user engagement metrics like clicks, likes, or shares. Further, these models evaluate images in isolation, without considering the contextual information about the image, such as company, time of releasing the image, *etc.*, reducing their effectiveness in predicting user engagement. This suggests that the existing metrics are not designed to capture the user engagement of images. Recent studies have employed the CLIP model as a proxy for human judgment (Nichol et al., 2022; Rombach et al., 2022), aiming to assess the alignment between generated images and text prompts. CLIP, trained on a diverse dataset, is thought to better capture nuanced aspects of human intention. However, similar to ImageReward and PickScore, the text prompts for CLIPScore come from image creators rather than users (viewers), which again is ineffective in conforming to viewers' expectations. The aesthetic

Table 2: Pearson correlation between model predicted scores and image engagement measured by account normalized likes.

| Model | Configuration | Pearson Correlation |
|---|---|---|
| PickScore (Kirstain et al., 2023) | - | 0.0734 |
| ImageReward (Xu et al., 2024) | - | 0.0285 |
| Human Preference Score (Wu et al., 2023) | - | 0.0747 |
| Aesthetic Score (Schuhmann et al., 2022) | - | 0.0674 |
| CLIP Score (Radford et al., 2021) | - | 0.0423 |
| GPT-3.5 (Ouyang et al., 2022) | 3-shot In-context learning | 0.0464 |
| GPT-3.5 (Ouyang et al., 2022) | 5-shot In-context learning | 0.0351 |
| GPT-4V (OpenAI, 2023) | 3-shot In-context learning | 0.1453 |
| GPT-4V (OpenAI, 2023) | 5-shot In-context learning | 0.1264 |
| EngageNet | Trained on random engagement score, Tested on actual engagement score | 0.0617 |
| EngageNet | Trained without MSE loss | 0.5821 |
| EngageNet | Trained without date input | 0.5365 |
| EngageNet | Trained without company input | 0.5226 |
| EngageNet | Trained without date and company input | 0.4476 |
| EngageNet | Trained without negative samples | 0.6051 |
| EngageNet | Trained with MSE loss and negative samples | 0.6248 |
| EngageNet | Oracle | 0.8682 |

score, built on pre-trained CLIP, is trained on several datasets capturing image aesthetics. However, as Fig. 1 shows, viewer engagement is much more nuanced than what aesthetics can capture.

Next, we try in-context learning with GPT-3.5 (Ouyang et al., 2022) and GPT-4-Vision (OpenAI, 2023) to predict viewer engagement over images. For GPT-3.5, we supply the image verbalization along with the Twitter handle and posting time, and for GPT-4-Vision, we give the actual image, the image verbalisation, the Twitter handle and posting time. We find that neither GPT-3.5 nor GPT-4 are able to predict user engagement accurately.

## 3.2 ENGAGENET MODEL TO ALIGN WITH VIEWER ENGAGEMENT

Since the existing approaches do not show acceptable performance for predicting user engagement over images, we, therefore, train our own engagement-aware vision-language model (VLM) model, EngageNet. To this end, we perform visual instruction fine-tuning of LLaVA-1.5 (Liu et al., 2023) on the EngagingImageNet train dataset (Figure 10). We design an instruction (Listing 1) for the VLM to predict the normalized likes of an image on a 0-100 scale, also conditioned on metadata comprising the marketer (company), image resolution, image colours and tones with their spatial coverage, image description and tags, and the date of releasing the image on social media.

We also augment EngagingImageNet with synthetic data samples. For this, we randomly sample 25% tweets from the high and medium likes buckets of each company and pair the tweet with an unrelated image from a different tweet. The corresponding likes is set to a low value, randomly sampled from the range 5-15. The resulting samples are called *negative samples*. This helps the model with the following: (1) it induces more sensitivity towards image features and reduces bias on tweet metadata to predict the engagement, and (2) penalises the image if it is irrelevant to the tweet context. We discuss this in more detail in Appendix F. Finally, we end up with a dataset of 957,809 samples, which we split into training and testing sets. We randomly sampled nearly 2000 samples from each bucket for testing, with the remaining samples used for training.

Since EngageNet is trained to predict the engagement score of an image, we additionally model the problem as a regression task. We attach a two-layered MLP network on top of the last layer of hidden states of the decoder module to predict the scalar engagement score from EngageNet. Therefore, while typically language models are trained on cross-entropy loss $L_{CE}$, we also use mean squared error $L_{MSE}$ as an auxiliary loss to train EngageNet. This is because $L_{MSE}$ is more sensitive to the difference between the predicted and actual engagement values, which is crucial to better guide EngageNet to learn the image engagement prediction task. Thus, the final loss function for EngageNet is given by:

$$L_{EOIG} = L_{CE} + \lambda L_{MSE} \tag{1}$$

where $\lambda$ is a hyperparameter that controls the weight of the auxiliary loss. We set $\lambda = 0.1$ in our experiments. We find that EngageNet demonstrates strong performance at predicting user engagement over images, achieving a Pearson correlation of 0.62 with ground truth user likes (Table 2).

We perform an ablation study to understand the impact of different components of the instruction on the performance of EngageNet. If EngageNet is not supplied with contextual information such as marketer company and the time of posting the image in the input, the correlation of its predictions with ground truth account normalized likes drops significantly (0.62 to 0.44). This indicates that the company and the time of posting are important components of the instruction for EngageNet to predict user engagement accurately. We also attempt to determine the impact of the auxiliary MSE loss adopted for training EngageNet. The MSE loss increases the correlation from 0.58 to 0.62, indicating that the auxiliary loss improves the performance of EngageNet. The MSE loss makes the model more sensitive to the difference between predicted and actual scores.

We also conduct an experiment to investigate the signal present in the EngagingImageNet dataset. For this, we train EngageNet on the EngagingImageNet dataset but with randomly sampled engagement values. We then evaluate the model on the EngagingImageNet test dataset with actual KPI values. In this case, the correlation of EngageNet's predictions with ground truth user likes is nearly zero, indicating that the engagement values in the EngagingImageNet dataset are crucial for EngageNet to learn predicting user engagement accurately.

Additionally, we evaluate the impact of adding negative samples to the training data. These samples are constructed by sampling images and other inputs in the instruction from different tweets, such that they are not aligned, and then setting the normalized likes to a very low value. Although we find that the addition of negative samples does not significantly impact the correlation of EngageNet, however it does help in improving the robustness of the model. This is because EngageNet learns to penalize images that are not aligned with the other inputs in the instruction. This is crucial for leveraging EngageNet as a reward model for engagement-optimized image generation as described in Section 4.3. Since we propose to also utilize EngageNet as an oracle for ranking models in the Engagement Arena, we train EngageNet on the entire EngagingImageNet dataset, *i.e.*, with both train and test data. In this configuration, EngageNet accomplishes a high correlation of 0.87 with ground truth user likes, which establishes its effectiveness to be used as an oracle.

### 3.3 GENERALIZATION CAPABILITY OF ENGAGENET

To assess the generalization ability of EngageNet across different social media platforms and engagement metrics, we evaluate its performance on the FlickrUser dataset (Abdullahu & Grabner, 2024). This dataset comprises 504,241 images from 2,337 users on Flickr, a popular photo-sharing platform. It includes engagement metrics such as the number of 'favorites' and number of 'views'.

From the full dataset, we use only 15,000 images for finetuning and a 5,000-image test set. As a baseline, we first fine-tune the pretrained LLaVA-1.5 Vision-Language Model (VLM) to predict the normalized 'favorites' (number of 'favorites' / number of 'views') as the engagement score of an image, leveraging contextual metadata such as upload date, description and tags. This model achieves a correlation of only 0.23 with the actual engagement scores in the test set.

Next, we finetune EngageNet for the same task. We find that despite being trained on minimal data, EngageNet significantly outperforms the baseline, achieving a correlation of 0.53 with actual image engagement scores. This substantial performance gain highlights EngageNet's ability to transfer its learned knowledge of image engagement to new contexts, demonstrating strong generalization across (1) different engagement metrics (favorites vs. likes), (2) alternative social media platforms (Flickr vs. Twitter), and (3) content types (enterprise vs. non-enterprise). These findings reinforce the robustness of EngageNet and its potential for broad applicability beyond its original training data. We also conduct a human study to assess the alignment between EngageNet's predicted engagement rankings and human preferences; detailed methodology and results are provided in Appendix C.

## 4 METHODS TO IMPROVE IMAGE ENGAGEMENT

We explore three methods for optimizing the text-to-image generation process to generate more engaging images. These include run-time and train-time optimizations: conditioning of text-to-image models on better prompts, supervised fine-tuning of stable diffusion on high-liked images, and reinforcement learning to align stable diffusion with EngageNet-based reward scores. The first method operates in the natural language domain at run-time, generating a description of how an engagement-optimized image should look like. On the other hand, the other two operate in the vision domain, generating actual engagement-optimized pixels by training the U-Net module of stable diffusion. We cover each of them next.

## 4.1 Conditioning Stable Diffusion on More Engaging Prompts

In the EngagingImageNet dataset, we observe that some images having similar themes but different details received vastly different levels of user engagement. For instance, consider the following pair of image captions: (1) "A living room having a couch and coffee table with a rug in front." (2) "A living room with large windows having a couch, coffee table and a rug" Despite both images depicting a similar scene (Figure 11), the first image received low engagement, while the second image garnered high engagement. In this case, the difference in engagement can likely be attributed to the presence of elements, such as large windows and natural light in the second image, which makes the living room appear bigger and more appealing to a viewer. Such observations motivated us to exploit patterns related to certain image aspects that can boost engagement. We further extend this analysis for images posted by a few companies in Appendix D.

Therefore, in this method (Figure 11), we attempt to alter the text prompts fed to the diffusion model such that the improved captions incorporate characteristics that have been empirically shown to boost image performance. For this, we adopt a retrieval framework described as follows. Using FAISS (Johnson et al., 2019; Douze et al., 2024), we index the vector embeddings of captions belonging to images in the high performance data subset of the EngagingImageNet train data as described in Section 2. Next, for every image caption in the low performance subset of the test data, we retrieve the semantically most similar caption from the corpus of high-performing images. If the similarity level is above a certain threshold $\tau$, the retrieved captions thus obtained are passed as input to the diffusion model for generating more performant images, otherwise the original caption is used for image generation.

## 4.2 Preferred Finetuning on High-Engagement Images

Several prior studies have demonstrated the feasibility of learning styles through stable diffusion by fine-tuning the model (Pinkney, 2022; Cjwbw, 2022; PromptHero, 2023; Everaert et al., 2023). These approaches typically involve fine-tuning the U-Net architecture within the Stable Diffusion framework using a set of images exhibiting the desired style. For instance, Everaert et al. (2023) proposed a method to finetune Stable Diffusion to adapt it to target styles like *anime sketches*, *American comics*, *Pokemon*, *starry night*, *etc.*

In this work, we attempt to explore whether the diffusion model can learn patterns associated with higher user engagement, analogous to learning visual styles. To this end, we performed fine-tuning of the base Stable Diffusion U-Net on the preferred data distribution, containing high liked images sampled from the EngagingImageNet train set (Figure 12). We call this process, **Preferred Finetuning**. The model was finetuned for 50 epochs, following the procedure outlined by von Platen et al. (2023). The model minimizes the standard denoising score matching loss (Ho et al., 2020; Ho & Salimans, 2022), which measures how well the model predicts the noise added to the image during the diffusion process:

$$L_{\text{denoise}} = \mathbb{E}_{\mathbf{x}_0, \boldsymbol{\epsilon}, t} \left[ \|\boldsymbol{\epsilon} - \boldsymbol{\epsilon}_\theta(\mathbf{x}_t, t, \mathbf{c})\|^2 \right] \tag{2}$$

Table 3: Results reveal the significant gains achieved in improving the engagement of low-liked subset of the EngagingImageNet dataset by enhancing the image descriptions fed to a text-to-image model, as described in Section 4.1.

| Images | Training Config | Oracle Engagement Reward | | Engagement reward increase |
|---|---|---|---|---|
| | | w/o prompt improvement | w/ prompt improvement | |
| DALL.E-2 | N.A. | 42.1459 | 45.7077 | 8.45% |
| SD 1.4 | N.A. | 38.7680 | 43.8173 | 13.02% |
| EOIG SD 1.4 | RLHF-ES | 40.2037 | 46.1218 | 14.72% |
| EOIG SD 1.4 | RLHF-DSG | 39.4950 | 44.8116 | 13.46% |
| EOIG SD 1.4 | Preferred Finetuning (PFT) | 43.1206 | 46.6490 | 8.18% |
| SD 1.5 | N.A. | 39.4001 | 44.3287 | 12.51% |
| SD 2.1 | N.A. | 45.4952 | 49.6946 | 9.23% |
| Pixart-alpha | N.A. | 44.7305 | 49.3322 | 10.29% |
| Pixart-sigma | N.A. | 46.2870 | 51.4671 | 11.19% |
| SD XL | N.A. | 49.1094 | 53.8602 | 9.67% |
| SD XL - DPO | N.A. | 51.3786 | 54.5863 | 6.24% |
| SD 3 Medium | N.A. | 50.6578 | 55.0841 | 8.74% |
| Flux.1-dev | N.A. | 48.7226 | 53.7209 | 10.26% |
| Ground Truth | N.A. | 41.2152 | 56.7533 | 37.70% |
| | | **Average Increase in Engagement** | | **12.41%** |

where $\mathbf{x}_0$ is the original image, $\mathbf{x}_t$ is the noisy image at time step $t$, generated by adding noise $\boldsymbol{\epsilon}$, $\boldsymbol{\epsilon}_\theta(\mathbf{x}_t, t, \mathbf{c})$ is the predicted noise from the model given the noisy image $\mathbf{x}_t$, time step $t$ and conditioning information $\mathbf{c}$ *i.e.*, text prompt fed as input to the diffusion model.

## 4.3 ALIGNING STABLE DIFFUSION WITH ENGAGEMENT

Black et al. (2023) proposed denoising diffusion policy optimization (DDPO), a policy gradient algorithm which frames the denoising process as a multi-step decision-making problem. The authors showed that DDPO can be employed to finetune text-to-image diffusion models to align their outputs with a variety of reward functions including image compressibility, aesthetic quality and image-prompt alignment, among others. Therefore, we explore the use of reinforcement learning to optimize diffusion models to improve the engagement potential of their generated images. To this end, we leverage EngageNet as a reward model to align a pre-trained stable diffusion model using DDPO algorithm to produce more engaging images. The entire process of alignment is shown in Figure 14 in the appendix. In DDPO, the denoising process is viewed as a finite horizon Markov decision process, where the state comprises of the current context, number of steps left in the process and the current denoised image. The action to be taken is to predict the next image using this state.

We experiment with two types of reward functions for finetuning stable diffusion:
(1) Engagement Simulation (ES): We leverage EngageNet to estimate the user engagement of images generated by stable diffusion. The reward signal is used to guide stable diffusion to generate higher engagement images as illustrated in Figure 14a. The resulting diffusion model is called EOIG-SD (RLHF-ES).
(2) Design Specification Generation (DSG): We train an alternate version of EngageNet to produce the design specification of an image, based on conditioning factors such as the company, time, image caption and viewer likes. This model learns to predict verbalized image descriptions comprising colors and tones with their spatial coverage, as well as objects with their locations, that should be reflected in an image, for a given engagement level and caption. The detailed method and results of EngageNet trained on this task are explained in Appendix J. Next, we utilise this EngageNet as a reward model to train stable diffusion such that the images generated by it have a design specification aligned with those of higher engagement images as shown in Figure 14b. EOIG-SD (RLHF-DSG) takes a text prompt and generates an image, which then undergoes verbalization via image perception models. Its objective is to create images that, when verbalized, closely resemble the engagement-conditioned verbalization generated by EngageNet. Thus, we ask EngageNet to provide the logits for this image verbalization, using which a reward is computed for EOIG-SD, indicating how closely this verbalized output aligns with EngageNet. This reward value serves as feedback for EOIG-SD in the form of policy gradient, aiding in its continual improvement and refinement within the image generation process. Only high engagement samples are used in the training process. The details of this method are described in Appendix K.

## 4.4 EVALUATING THE METHODS ADOPTED FOR ENGAGEMENT-OPTIMIZATION

**Run-time optimization:** Firstly, we investigate the impact of using better prompts to condition the text-to-image generation process as described in Section 4.1. The results are summarised in

Table 4: Comparing the performance gains on the EngagingImageNet test dataset, resulting from train-time engagement-optimization methods applied on stable diffusion, as described in Sections 4.2 and 4.3.

| Images | Training Config | Bucket | Engagement Reward | Engagement Increase | Aesthetic Score | CLIP Score | FID | PickScore |
|---|---|---|---|---|---|---|---|---|
| Ground Truth | N.A. | High | 90.9526 | N.A. | 5.1006 | 32.6343 | N.A. | 20.9470 |
| | | Medium | 74.3535 | N.A. | 5.0940 | 32.4867 | N.A. | 20.9351 |
| | | Low | 41.2152 | N.A. | 5.0518 | 32.2012 | N.A. | 20.7406 |
| SD 1.4 | N.A. | High | 56.1489 | N.A. | 5.2029 | 33.0830 | 24.6631 | 17.3514 |
| | | Medium | 51.6949 | N.A. | 5.1634 | 32.9173 | 23.4434 | 17.3344 |
| | | Low | 38.7680 | N.A. | 5.1662 | 32.8339 | 24.2607 | 17.3262 |
| EOIG SD 1.4 | Preferred Finetuning (PFT) on High engagement Images | High | 62.0390 | 10.49% | 4.8090 | 32.4524 | 24.9370 | 17.3070 |
| | | Medium | 56.1082 | 8.54% | 4.8387 | 32.3923 | 23.0904 | 17.3239 |
| | | Low | 43.1206 | 11.23% | 4.8108 | 32.2960 | 23.5885 | 17.2932 |
| EOIG SD 1.4 | RLHF - ES | High | 58.2724 | 3.78% | 5.1828 | 33.2891 | 23.8656 | 17.4113 |
| | | Medium | 53.1004 | 2.72% | 5.1686 | 33.2845 | 22.7157 | 17.3802 |
| | | Low | 40.2037 | 3.70% | 5.1629 | 32.9468 | 24.1780 | 17.3672 |
| EOIG SD 1.4 | RLHF - DSG | High | 57.9188 | 3.15% | 5.2495 | 33.1072 | 23.9144 | 17.3577 |
| | | Medium | 52.9765 | 2.48% | 5.2187 | 33.0991 | 23.3626 | 17.3486 |
| | | Low | 39.4950 | 1.88% | 5.2336 | 32.9716 | 24.2147 | 17.3277 |

Table 3. By retrieving semantically similar captions from the corpus of high-liked images, visual characteristics that have been empirically shown to enhance image engagement get incorporated in the text prompt. Therefore, after applying this method, we observe a significant improvement in the engagement of low-liked subset of the EngagingImageNet test dataset, consistent across multiple text-to-image models, both open-source and closed-source. This method is highly effective as it is able to produce images with higher engagement without any additional training of the diffusion model. We observe that, on average, an improvement in the prompt results in an improvement of 12.4% in engagement. The improvement is observed in models across all sizes and also for models trained on high-engagement images (EOIG-SD).

**Train-time optimization:** Next, we present the results of the methods (§4.2, §4.3) adopted for engagement-optimized image generation by training the U-Net module of stable diffusion in Table 4. We denote all the models trained using train-time optimizations like Preferred fine-tuning with EOIG (engagement optimized image generation). We compare the performance of the base stable diffusion model (SD 1.4), stable diffusion finetuned on high-engagement images (EOIG-SD PFT), and stable diffusion aligned with EngageNet-based reward functions (EOIG-SD RLHF-ES, EOIG-SD RLHF-DSG). For this, we use EngageNet-Oracle as a judge to predict the user engagement of the images generated by these models. This helps us probe the effect of different training strategies on improving the engagement capabilities of stable diffusion. Consistent with prior literature, we also include other metrics like FID (Heusel et al., 2017), aesthetics (Schuhmann et al., 2022), CLIP score (Radford et al., 2021), and PickScore (Kirstain et al., 2023).

The results indicate that while all the training methods improve the engagement capabilities of stable diffusion, however, the extent of improvement varies widely. We find that finetuning the stable diffusion model on preferred data distribution, i.e. high-engagement samples from the EngagingImageNet dataset yields significant gains in the engagement potential of the generated images. This is evident from the consistent increase in the predicted user engagement of the generated images across all engagement buckets. Next, we discover that using EngageNet-based reward functions to align the stable diffusion model also results in better performance. However, the improvement in image engagement is not as significant as that achieved by the previous methods. Other metrics like CLIP score, PickScore and FID do not vary significantly across the EngagingImageNet buckets and largely remain unaffected after training stable diffusion in both the above regimes. This further corroborates their non-alignment with image engagement. We also discuss the results of applying train-time optimizations on other text-to-image models in Appendix I.

Next, we discuss the side effects of training stable diffusion using the above methods. As a consequence of training on engaging images, we find that stable diffusion learns to generate images with certain persuasion strategies (Kumar et al., 2023). For instance, Fig. 13 shows several examples of product and model photography generated by EOIG-SD and base SD, demonstrating EOIG-SD's biases towards certain persuasion strategies such as social appeal and social identity, commonly observed in marketing scenarios (Kumar et al., 2023) but ignored in general photography.

**Combination of Train-time and Run-time optimizations:** In our experiments, we gauge the impact of different methods in improving image engagement by comparing the results of both train-time (§4.2, §4.3) and run-time (§4.1) optimizations, as well as their combination. Stable Diffusion 1.4 (Rombach et al., 2022), serves as the baseline model. In Table 3, we observe that when each method is applied individually, such as using better prompts at run-time or training the diffusion model through supervised finetuning or using reinforcement learning, it results in measurable improvements in image engagement over the baseline. However, the most significant improvements are seen when supervised finetuning or reinforcement learning is combined with better prompts at run-time. This demonstrates that coupling train-time and run-time optimizations has a synergistic effect, resulting in higher engagement levels than each method applied alone.

## 5 *Engagement Arena*: MEASURING ENGAGEMENT CAPABILITIES OF TEXT-TO-IMAGE MODELS

Motivated by the work of LMSYS and similar benchmarks (Chiang et al., 2024), we propose *Engagement Arena* as a platform to evaluate the capability of text-to-image models to generate engaging images. We run a tournament on a common set of prompts from the EngagingImageNet test set. We leverage EngageNet as an oracle for *Engagement Arena* to compute the Elo ratings of various

open-source text-to-image models, such as Stable Diffusion 3 Medium (Esser et al., 2024), Flux.1-dev (Labs, 2024), Stable Diffusion XL (Podell et al., 2023), Stable Diffusion XL-DPO (Wallace et al., 2024), Pixart-sigma (Chen et al., 2024a), Pixart-alpha (Chen et al., 2024b), Stable Diffusion 2.1, Stable Diffusion 1.5, Stable Diffusion 1.4, (Rombach et al., 2022), *etc.*, and closed-source models like DALL.E-2 (Ramesh et al., 2022). Figure 3 shows the rankings of these models. It also features the Elo ratings of ground truth images to serve as topline for benchmarking the models.

In addition to helping to rank the engagement potential of generated images accurately, using EngageNet as an oracle also avoids having static benchmarks with a definitive ground truth. We encourage the research community to adopt *Engagement Arena* as a basis for measuring further advances in the engagement capabilities of text-to-image modeling and incorporating user engagement into the learning process.

The arena features actual images from the EngagingImageNet dataset as a topline benchmark for the images generated by different text-to-image models. We find that Stable Diffusion 3 Medium (Esser et al., 2024) emerges as the best performing model in the *Engagement Arena*, with a win rate of 46% over actual images (Figure 15). It is followed by SDXL-DPO (Wallace et al., 2024) and Flux.1-dev (Labs, 2024). We notice a general trend that image engagement rises with the size of the text-to-image models. However, there are some exceptions to this trend. For instance, Pixart family of models (600M parameters) and EOIG-SD PFT model (860M parameters) surpass relatively larger DALL.E-2 (6.5B parameters).

We observe that while our EOIG-SD models trained using different methods (PFT, RLHF) outperform the equal-sized base SD 1.4 model, however there is a considerable gap between the performance of EOIG-SD and significantly larger text-to-image models leading the arena. This can be attributed to the inherent limitations of SD 1.4 in generating high-quality images, which cannot be fully overcome by the training methods explored in this work.

## 6 CONCLUSION

As text-to-image generation models continue to evolve, their success must be measured not only by aesthetic quality and realism but also by their ability to drive meaningful engagement. In this work, we introduced the challenge of optimizing image generation for engagement, a crucial factor in e-commerce and advertising. Our findings reveal that traditional image evaluation metrics, such as aesthetics and CLIPScore, fail to correlate with user engagement, necessitating new approaches. To address this, we curated EngagingImageNet, a large-scale dataset that captures real-world user engagement over images, and developed EngageNet, a vision-language model designed to predict user engagement. By leveraging EngageNet as a reward function, we explored multiple strategies to enhance engagement in text-to-image models, including prompt optimization, fine-tuning on high-engagement images, and reinforcement learning alignment with engagement-based rewards. These methods demonstrate promising improvements in generating images that resonate more with viewers. Furthermore, we introduced the Engagement Arena, a benchmarking platform that enables the research community to evaluate and compare text-to-image models based on engagement performance. We provide a detailed discussion of broad directions, along with promising future directions for real-world applications in Appendix B.

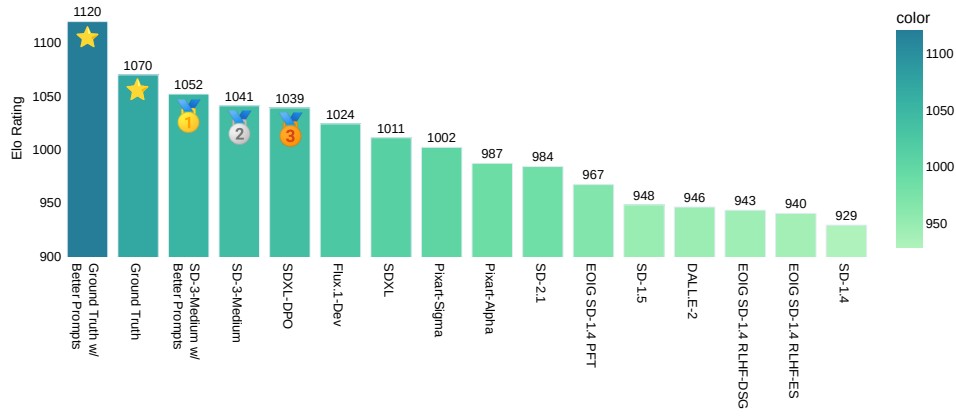

Figure 3: Rankings and Elo ratings of various text-to-image models in the proposed Engagement Arena.

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

APPENDIX

# A    RELATED WORK

## A.1    TEXT-TO-IMAGE GENERATION

A large body of work has been done with respect to generating images from textual descriptions. These text controlled image generation models have evolved greatly from the time of GANs (Goodfellow et al., 2020) to yield high-quality image generators based on diffusion models such as DALL-E (Ramesh et al., 2021), Stable Diffusion (Rombach et al., 2022) and ones that have extended these models, that are able to follow human text instructions to a large extent. However, the metrics that these generators optimize are Inception Score (IS) (Salimans et al., 2016) and Fréchet Inception Distance (FID) (Heusel et al., 2017). It has been observed by multiple works for example (Kirstain et al., 2023; Wu et al., 2023; Xu et al., 2024) etc. that these metrics do not necessarily correspond to human preferences.

## A.2    ALIGNING IMAGE GENERATION WITH HUMAN PREFERENCES

To align models with human preferences, reinforcement learning with human feedback (RLHF) has been successfully used in the LLM literature (Ouyang et al., 2022; OpenAI, 2023; Touvron et al., 2023) with algorithms such as PPO (Schulman et al., 2017), DPO (Rafailov et al., 2024) and several variants of these preference based reinforcement learning algorithms.

Similar approaches have also been used in text-to-image generation models. In these approaches, the latent image generation part of the diffusion model (either UNet or a Transformer) is trained using a reward model in the case of DDPO (Black et al., 2023) or using user preferences directly in the case of DPO (Wallace et al., 2024). Both these approaches involve collecting human preference datasets.

Liang et al. (2024) introduce a dataset RichHF-18K and a multimodal transformer model (RAHF) that aims to predict detailed human feedback, such as identifying implausible image regions and misaligned text. They collected rich human feedback over 18,000 images from various users. Then they tune the RAHF model to predict implausibility scores, heatmaps of artifact locations, and text-image misalignment. The predicted feedback is utilized to enhance image generation quality through targeted finetuning.

Du et al. (2024) curate a large-scale dataset consisting of over one million generated advertising images, each annotated with human feedback regarding their suitability for advertising purposes. This dataset is created using a diverse selection of products from JD.com, featuring generated advertising images alongside corresponding product images with transparent backgrounds and meticulously crafted prompts by professional designers. Additionally, the paper introduces a multimodal model trained on this dataset to simulate human feedback of advertising suitability and automate image evaluation. This model utilizes an efficient recurrent generation process to produce high-quality advertisement images.

Clark et al. (2024) propose a method for gradient-based reward finetuning based on differentiation in the diffusion sampling process. This approach, Direct Reward Fine-Tuning (DRaFT) allows for optimising diffusion models by incorporating differentiable rewards based on human feedback. They demonstrate the application of this method on a variety of reward functions, such as aesthetic score (Schuhmann et al., 2022), PickScore (Kirstain et al., 2023) and Human Preference Score (Wu et al., 2023).

Liang et al. (2024); Du et al. (2024); Clark et al. (2024) use simulated human feedback to refine the image generation process for specific image quality dimensions of realism (Liang et al., 2024; Du et al., 2024; Clark et al., 2024), aesthetics (Du et al., 2024; Clark et al., 2024), visual appeal (Clark et al., 2024) or advertising suitability (Du et al., 2024).

## A.3    DATASETS AND METRICS FOR HUMAN PREFERENCES OVER IMAGES

Several human preference datasets for text-to-image generation have been collected in literature. These include Pick-a-Pic dataset (Kirstain et al., 2023), dataset generated from the Stable Foundation

Discord channel (Wu et al., 2023), ImageReward dataset (Xu et al., 2024), etc. The human preferences in these datasets have been collected by explicitly asking humans to state their choices. These datasets are often accompanied with their own metrics for human preference alignment such as PickScore (Kirstain et al., 2023), Human Preference Score (Wu et al., 2023), ImageReward score (Xu et al., 2024), etc. While the authors of these research works have shown that these metrics are better aligned with human preferences when compared with conventional metrics such as CLIP score, BLIP score, Aesthetic Score, etc, however, they still need not align with viewer engagement of images. In this work, viewer engagement refers to the measurable interactions a viewer exhibits toward an image on a digital platform, reflecting how strongly the content resonates and elicits actionable behavior. We leverage implicit human preference signals derived from engagement metrics, specifically account-normalized likes in the EngagingImageNet dataset, as a proxy for viewer engagement.

Thus, we explore holistic and user-centric metrics of engagement that measure dimensions beyond traditional image quality. Our contribution is unique in the sense that we prioritize viewer engagement as the core feedback, allowing for engagement-optimized image generation that directly aligns with how users interact with visual content online. While traditional metrics like aesthetics, FID, ImageReward, image realism help in making the images look real, aesthetic and structurally correct, optimizing and measuring viewer engagement of images has potential applications in personalization, efficient A/B testing, recommender systems, and even public awareness and information campaigns as discussed in Appendix B.

## B  How To Contribute Towards The Goal Of Engagement Optimization And Directions For Future Research

Building on the foundational work of this paper, future research could explore the following areas:

- Project Website: Our project website (behavior-in-the-wild.github.io/image-engagement) shall serve as a platform for researchers interested in engagement optimization for text-to-image models. We will release the following artifacts: (1) EngagingImageNet dataset, (2) Model training code, and (3) Engagement Arena for benchmarking text-to-image models. We envision it as an open-source platform, inspired by initiatives like LMSys (https://lmsys.org/) and LabintheWild (https://labinthewild.org/).
  We welcome contributions in several forms:
  - Adding new engagement-labeled image datasets from diverse sources
  - Conducting and contributing results from user studies that further advance the understanding in this field
  - Running engagement evaluations on newly developed text-to-image models and contributing results to EngagementArena
  - Assisting in platform maintenance and improvement
  - Donations for supporting with infrastructure and computational resources
- Human Studies: Conducting new human studies can further advance the understanding of engagement dynamics across platforms and user behaviors. Some key research questions include how engagement patterns vary across platforms, the relationship between user characteristics (e.g., social media usage) and engagement, and the distinction between short-term campaign engagement (e.g., likes, click-throughs) and long-term user retention or loyalty. Investigating these factors can help refine engagement prediction models, improve EngageNet's adaptability, and provide deeper insights into optimizing content for diverse audiences.
- Adapting EngageNet for diverse engagement signals: As demonstrated in the Flickr dataset experiment in Section 3.3, finetuning EngageNet on minimal data led to strong performance gains in predicting image engagement. This highlights EngageNet's ability to transfer its learned knowledge of image engagement to new social media platforms and engagement metrics. Similarly, researchers can further fine-tune EngageNet on diverse datasets to model engagement dynamics across platforms such as Instagram, YouTube, and TikTok, capturing a broader spectrum of user behaviors. To support such efforts, we have provided our fine-tuning code on the project website.

- Impact of External Factors on Engagement: In the current work, potential unobserved confounders such as external events or viral trends, are not explicitly accounted for, despite their potential impact on engagement metrics. These factors can introduce variations in user interactions that are not directly modeled. Future research can explore approaches to identify and mitigate the influence of such external factors, improving the robustness of engagement prediction models.

- Explaining Content Factors Leading to Higher Engagement: Understanding why certain visuals generate higher engagement is crucial for both creators and researchers. Future work could involve enhancing EngageNet and Engagement Arena to capture subtle engagement dimensions such as emotional resonance, visual complexity, and long-term memorability.

- Prescribing Content Strategies to Creators: Beyond explaining engagement factors, future iterations of EngageNet could actively prescribe actionable strategies to creators. For instance, it could suggest specific attributes (e.g., color schemes, composition, or visual themes) that are likely to increase engagement for a particular audience or context, thus empowering creators to produce high-performing content efficiently.

- Video Engagement Optimization: Extending the proposed framework to handle video content could provide insights into optimizing video advertisements, storytelling, and social media clips for engagement. It involves considering dynamic elements like pacing, transitions, and viewer retention metrics, which play a critical role in video content performance.

- Fashion Industry: The fashion industry could benefit from integrating engagement-aware content generation into virtual try-on systems. Models could generate visuals tailored to consumer preferences, encouraging them to explore and interact with virtual products. This could also enhance online shopping experiences by presenting customers with styles predicted to resonate with their personal tastes.

- Enhanced Recommender Systems: Engagement-aware images have the potential to enhance recommendation models significantly. By integrating these visuals into recommender systems across domains like retail, food delivery, and streaming services, businesses can present more appealing and relevant suggestions to users, thereby boosting engagement and conversions.

## B.1    BROADER APPLICATIONS

While this work primarily focuses on optimizing text-to-image models for engagement in commercial and marketing settings, the proposed EngageNet framework has broader implications across various domains:

- **Personalization, Campaign Optimization and Recommender Systems**: The EngageNet model can assist marketers and advertisers in selecting images that maximize viewer engagement, enabling automatic asset selection for brand campaigns. By integrating with multimodal recommendation systems, EngageNet can enhance the relevance and appeal of suggested images in applications such as personalized content feeds and digital advertising. This makes EngageNet a valuable tool for valuable tool for campaign optimization and automated content curation.

- **Health and Public Awareness Campaigns**: Generating compelling visuals for public awareness campaigns, such as health education and environmental initiatives, can significantly amplify their effectiveness. Research in public health and medicine highlights the positive outcomes of interventions like engaging advertisements on audience behavior (Wakefield et al., 2010; Kite et al., 2023). By leveraging EngageNet, organizations can optimize imagery to enhance message retention and public engagement.

- **Infinite Personalization**: Currently, ad delivery engines choose from a fixed pool of creative assets for each visitor. With advances in generative AI, especially LLMs and diffusion models, it is now possible to create multiple ad variations dynamically. EngageNet can score these variants in real-time for each visitor, enabling infinite personalization, where every individual receives bespoke content, maximizing relevance and engagement.

- **Efficient A/B Testing**: A/B testing is a cornerstone of optimizing marketing campaigns. However, due to the complexity and resource-intensive nature of A/B tests, few marketers are actually able to use them. Among those who engage in A/B testing, the data-intensive requirements necessary to reach statistical significance often limit them to testing only a few

variants at a time. Thus, marketers can employ EngageNet to shortlist and prioritize images with high engagement potential, leading to more efficient and data-driven A/B experiments.

## C  STUDY TO EVALUATE ALIGNMENT OF ENGAGENET WITH HUMAN PREFERENCES

We conduct a human study to validate how well the automated rankings produced by EngageNet correspond to human-perceived engagement. Through a human study done with marketers, Singh et al. (2024) found that the judgment of expert marketers has little correlation with advertisement success, thus validating the need to have an automated metric to measure engagement. Next, we conduct a human study to help us analyze how much the automated rankings produced by EngageNet align with humans. We describe the procedure and results for the human study below.

### C.1  EXPERIMENT DESIGN AND METHODOLOGY

From the Engagement Arena, we select three text-to-image models with sufficiently different Elo ratings to ensure clear differentiation in engagement rankings: (A) Stable Diffusion-3-Medium (Elo=1041), (B) Pixart-Sigma (Elo=1002), and (C) DALL·E-2 (Elo=946). We use images generated by these models to construct pairwise comparisons for the study, described as follows.

The study consists of a survey-based human evaluation, divided into two parts:

- Part A - Advertising and Social Media Exposure: Participants answer a short questionnaire about their advertisement consumption habits and social media exposure to establish familiarity with digital media.
- Part B - Pairwise Image Comparisons: Participants are shown 60 image pairs, each generated from the same text description by two different models. They select the image they find more visually appealing and engaging. Each participant evaluates 60 randomly sampled image pairs from a pool of 200 pairs, divided into 20 comparisons per model pair, ensuring a balanced distribution for comparing different models with each other.

### C.2  RESULTS

For each pairwise comparison, we check whether the human-selected image matches EngageNet's predicted ranking. If the rankings are identical, the comparison is deemed aligned.

The average alignment score is computed as:

$$\text{Average Alignment Score} = \frac{\text{Number of aligned comparisons}}{\text{Total comparisons across all users and models}} \quad (3)$$

Based on the current results, the average alignment score is 64.41%, indicating good correlation between human preferences and EngageNet-predicted rankings. The study is continuously collecting samples and updating its results. To take the study and to see more detailed analysis, please visit our project webpage: `behavior-in-the-wild.github.io/image-engagement`.

## D  ANALYSING VISUAL ASPECTS THAT DRIVE ENGAGEMENT

To understand the visual aspects that often lead to higher image engagement, we analysed pairs of images having same theme but different details, posted within a 45 days interval by the same account. Then pairs with vastly different engagement levels between the images were sampled. We then extracted the differences between the image pairs for a few companies using using GPT-4-Vision (OpenAI, 2023). Following are some main observations. For fashion brands like Bulgari, we observe that images featuring prominent branding and dynamic backgrounds with bright colors and gradients significantly enhance engagement, as visible in Figure 1 (Image pair-3). For Gucci, engagement is driven by images that maintain a clear focus on the product, emphasizing intricate detailing and textures. Additionally, images that incorporate luxurious backgrounds contribute to higher engagement levels. In the case of Airbnb, images that blend natural light with greenery are particularly effective in enhancing user engagement. Showcasing relatable homestay experiences

aligns closely with Airbnb's branding, further driving engagement. Meanwhile, Lenovo benefits from highlighting unique technical features and specifications while utilizing vibrant colors and high-contrast backgrounds.

## E    ENGAGINGIMAGENET FILTERING STEPS

We sample the tweets posted in the 5 year time period from January 2018 to January 2023. We focus our analysis on usernames that market products or services, and thus weed out usernames belonging to categories like news and sports. Next, we check if the number of tweets posted by a username exceeds 1000, then we retain the username, else we discard remove it. This helps in removal of stray handles and ensures data quality. Further, if the number of tweets posted by a username exceeds 2000, we randomly sample 2000 tweets for this username to avoid oversampling tweets from the same username and thus compromising data variance. This step ensures that the dataset is fairly representative of different enterprise accounts. Moreover, we weed out all tweets containing media other than images and where tweet text is less than 50 characters. Also, the hyperlinks present in the tweets are masked with a <hyperlink> placeholder. This results in 365,129 tweets posted in 5 years by 592 Twitter handles. Since it is hard to assign engagement (likes) credit to the multiple media present in a single tweet, we assign an equal engagement credit to all the media in a tweet.

## F    NEGATIVE SAMPLES FOR ENGAGENET FINETUNING

Negative samples were introduced to enhance EngageNet's ability to discriminate between relevant and irrelevant image-text pairs, improving its overall robustness. The method for constructing such misaligned pairs is as described below. We augment EngagingImageNet with synthetic data samples. For this, we randomly sample 25% tweets from the high and medium likes buckets of each company and pair the tweet with an unrelated image from a different tweet. The corresponding normalized likes is set to a low value, randomly sampled from the range 5-15. The resulting samples are called negative samples. This process ensured that the negative samples presented a challenging scenario for EngageNet, where the image and textual context were semantically misaligned. Thus, adding negative samples in training EngageNet is conceptually analogous to adding negative samples in contrastive learning.

This helps the EngageNet model with the improved sensitivity to contextual alignment of images. Training on negative samples compelled EngageNet to pay closer attention to the semantic relationship between an image and its contextual tweet metadata. EngageNet learns to penalise an image if it is irrelevant or poorly aligned to the tweet context. This ability is crucial to utilize EngageNet as an oracle for evaluation of text-to-image models in the Engagement Arena, such that aesthetically pleasing but irrelevant generated images do not score high on engagement.

## G    ENGAGINGIMAGENET ADDITIONAL DETAILS

Table 5: Distribution of ground truth EngagingImageNet images

| Engagement Level | # Objects | Aesthetic Score | CLIP Score |
|------------------|-----------|-----------------|------------|
| High | 3.405 | 4.994 | 30.509 |
| Low | 3.291 | 4.881 | 30.638 |

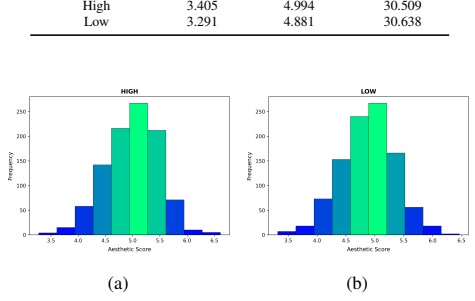

Figure 4: Aesthetic Score distribution across High and Low engagement images in EngagingImageNet dataset

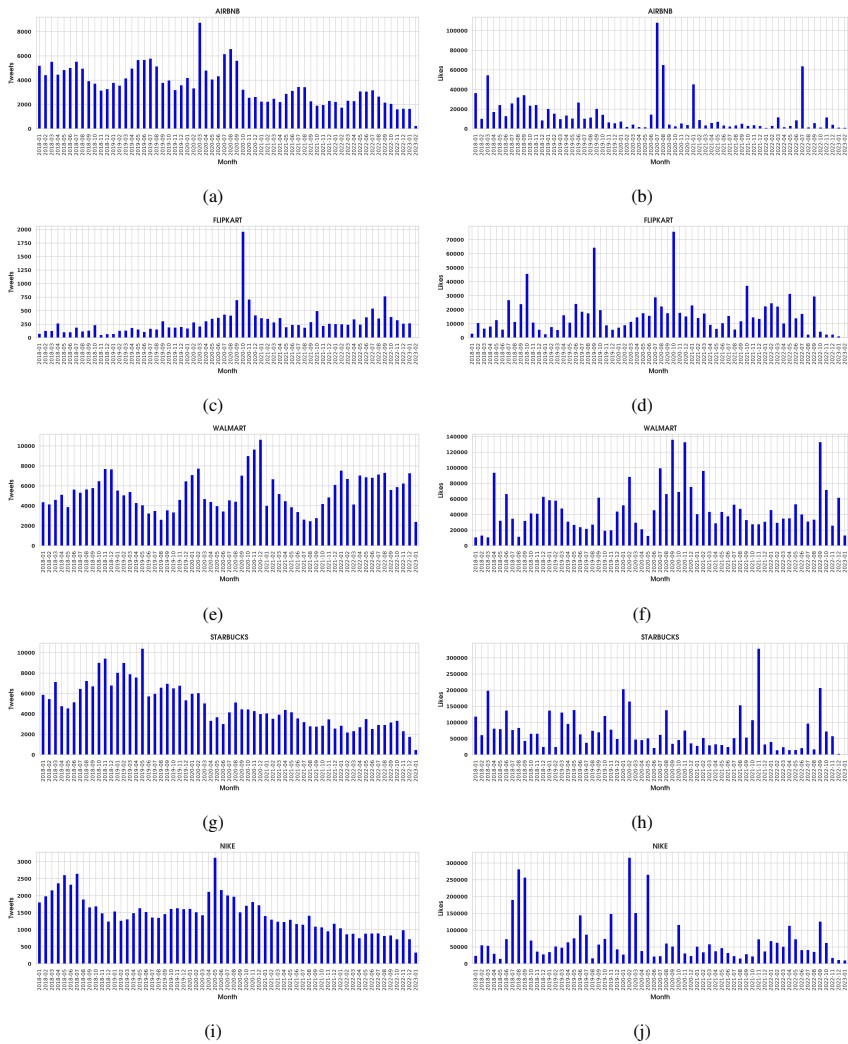

Figure 5: Plots showing variation of number of tweets and likes with time for a few companies in the EngagingImageNet dataset

# H    PROMPTS FOR INSTRUCTION FINETUNING

Listing 1: Visual instruction finetuning Pattern: EngageNet predicts user engagement of images given contextual information about the social media post.

```
Input : <image>
This is an image that a marketer from company "gucci" wants to post on social media for marketing purposes. The following
        information about this image is also given :
(1) image resolution i.e. (width, height): [680, 680],
(2) image colors and tones: {"color and tones": {"colors": {"Orange": {"coverage": 0.6}, "White": {"coverage": 0.18}, "Pink":
        {"coverage": 0.12}, "Brown": {"coverage": 0.1}}, "tones": {"warm": 0.72, "neutral": 0.28, "cool": 0}}},
(3) marketer's intended image description : A girl with a nose ring and gold earrings .,
(4) marketer's intended image tags: nose ring, gold earrings, girl, makeup, lips, face, beauty, earrings, nose, lips, gold,
        woman, makeup, accessories,
(5) date of posting : 22-February-2019
Now, carefully observe the image. You have to predict the "number of likes" that this image will get, on a scale of 0 to 100.
It measures the number of times the viewers will interact with the social media post by clicking the "Like" button to express
        their appreciation for the image. Thus, an image with higher visual appeal, alignment with the company's brand identity
        , and relevance to the audience, is likely to receive more likes. Moreover, a good image should stongly correspond with
         the marketer's intended image description and tags to attract the target audience.
Your predicted "number of likes" will help the marketer to decide whether to post this image or not on the social media
        platform.
Answer properly in JSON format. Do not include any other information in your answer.
```

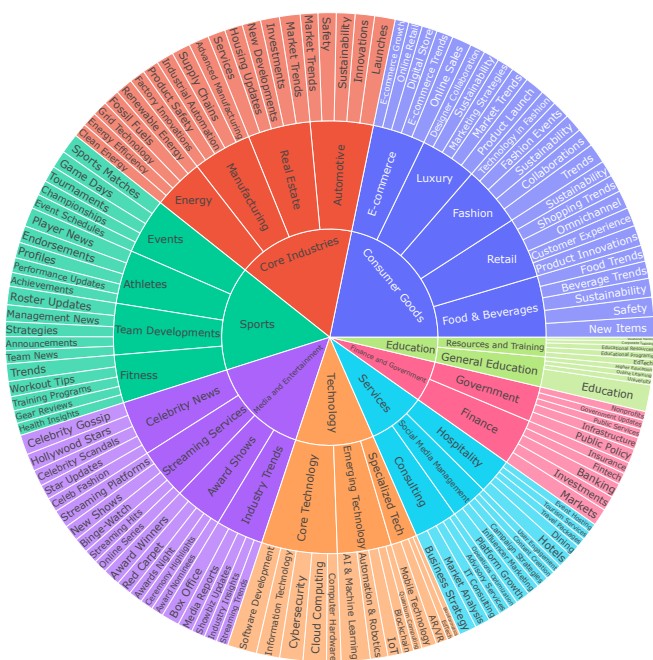

Figure 6: To analyze the distribution of tweet topics, we used BERTopic to extract topics from the tweets. These topics were subsequently clustered and named using GPT-4o-mini. This figure demonstrates that EngagingImageNet data encompasses a wide range of diverse tweet topics.

```
Output:
{"likes": 19}
```

Listing 2: Engagement Finetuning Verbalization Pattern (1): Explicitly asking model to pay attention to engagement tokens

```
Input: You are a smart model. I am giving you some data regarding an image – (1) captions (2) keywords (3) image resolution i.
    e. (width, height) (4) release date (5) number of downloads i.e. how many times the image was downloaded (6) number of
    forwards i.e. how many times the image was forwarded to someone else (7) number of impressions i.e. how many times the
    image was seen by someone. Note that (5), (6) and (7) are Key Performance Indicators (KPIs) of the image, thus they are
    important signals of its perceived quality and popularity.
You have to predict following attributes of the image: (1) colour and tones from the lists given below: – Allowed colours: ['
    Red', 'Dark_Red', 'Green', 'Bright_Green', 'Dark_Green', 'Light_Green', 'Mud_Green', 'Blue', 'Dark_Blue', 'Light_Blue',
    'Royal_Blue', 'Black', 'White', 'Off_White', 'Gray', 'Dark_Gray', 'Silver', 'Cream', 'Magenta', 'Cyan', 'Yellow', '
    Mustard', 'Khaki', 'Brown', 'Dark_Brown', 'Violet', 'Pink', 'Dark_Pink', 'Maroon', 'Tan', 'Purple', 'Lavender', '
    Turquoise', 'Plum', 'Gold', 'Emerald', 'Orange', 'Beige', 'Lilac', 'Olive'] – Allowed tones: ['warm', 'neutral', 'cool
    '] (2) main objects present in the image and the diagonal coordinates of their bounding boxes: [x1, y1, x2, y2]
Now, predict the attributes for the following image: [captions: "Waist up portrait of mixed–race female worker posing
    confidently while standing with arms crossed in plant workshop", keywords: "female, worker, young, woman, mixed–race,
    african, african–american, modern, contemporary, work, occupation, industry, industrial, plant, factory, workshop, work
    shop, strong, tough, gritty, masculine, short, hair, latin–american, plump, adult, mechanic, repair, repairman,
    handywoman, foreman, copy space, portrait, looking at camera, standing, posing, smiling, recruitment, employment, job,
    opportunity, engineer, production, manufacturing, assembly, assembling, line ", image resolution: "(5760, 3840)",
    release date: "2019–12–02", number of downloads: "24", number of forwards: "106", number of impressions: "5941"]
    Answer properly in JSON format. Do not include any other information in your answer.

Output: {"color and tones": {"colors": {"Gray": {"coverage": 0.4}, "Dark_Gray": {"coverage": 0.22}, "Black": {"coverage":
    0.14}, "Off_White": {"coverage": 0.13}, "Silver": {"coverage": 0.11}}, "tones": {"warm": 0, "neutral": 1.0, "cool":
    0}}, "objects": {"jeans": [2076.67, 2542.5, 3023.88, 3827.01], "woman": [1892.94, 11.18, 4260.09, 3824.34], "safety
    vest": [2160.75, 1410.95, 3668.16, 3826.63], "shirt": [2163.59, 1079.3, 4254.59, 3826.1]}}
```

Listing 3: Engagement Finetuning Verbalization Pattern (2): Noisy engagement in input and asking the model to correct the noise in addition to producing content

```
Input: "You are a smart model. I am giving giving you some data regarding an image released by a content creator – (1)
    captions (2) keywords (3) image resolution i.e. (width, height) (4) release date (5) approximate number of downloads
    that the creator wants to achieve (6) approximate number of forwards that the creator wants to achieve (7) approximate
    number of impressions/views that the creator wants to achieve
You have to predict following attributes of the image: (1) colour and tones from the lists given below: – Allowed colours: ['
    Red', 'Dark_Red', 'Green', 'Bright_Green', 'Dark_Green', 'Light_Green', 'Mud_Green', 'Blue', 'Dark_Blue', 'Light_Blue',
```

```
    'Royal_Blue', 'Black', 'White', 'Off_White', 'Gray', 'Dark_Gray', 'Silver', 'Cream', 'Magenta', 'Cyan', 'Yellow', '
    Mustard', 'Khaki', 'Brown', 'Dark_Brown', 'Violet', 'Pink', 'Dark_Pink', 'Maroon', 'Tan', 'Purple', 'Lavender', '
    Turquoise', 'Plum', 'Gold', 'Emerald', 'Orange', 'Beige', 'Lilac', 'Olive'] - Allowed tones: ['warm', 'neutral', 'cool
    '] (2) main objects present in the image and the diagonal coordinates of their bounding boxes: [x1, y1, x2, y2] (3)
    exact number of downloads that the image will get  (4) exact number of forwards that the image will get  (5) exact
    number of impressions/views that the image will get
Now, predict the attributes for the following image: [ captions: ""Hispanic adult man holding 100 brazilian real banknotes
    smiling happy pointing with hand and finger to the side "", keywords: ""pointing, side, face, happy, hopeful, smile,
    finger, optimistic, hand, point, showing, looking, smiling, one, gesture, confident, up, cheerful, look, mouth, joy,
    friendly, expression, emotion, presentation, idea, blue, background, hispanic, latin, man, male, guy, beard, bald,
    shaved, adult, young, money, currency, business, brazilian, cash, brazil, real, investment, banknote, 100"", image
    resolution: ""(9216, 6144)"", release date: ""2021-02-27"", approximate number of downloads that the creator wants to
    achieve: ""4"", approximate number of forwards that the creator wants to achieve: ""17"", approximate number of
    impressions/views that the creator wants to achieve: ""919"" ] Answer properly in JSON format. Do not include any other
    information in your answer."

Output: {"color and tones": {"colors": {"Cyan": {"coverage": 0.69}, "Light_Blue": {"coverage": 0.16}, "Turquoise": {"coverage
    ": 0.15}}, "tones": {"warm": 0, "neutral": 0, "cool": 1.0}}, "objects": {"man": [1068.6, 18.57, 8143.44, 6121.09], "
    banknote bill": [3443.5, 2146.79, 5294.15, 3455.05]}, "exact downloads": 4, "exact forwards": 15, "exact impressions":
    885}
```

Listing 4: Engagement Finetuning Verbalization Pattern (3): Noisy engagement in input and asking the model to correct engagement in output

```
Input: "You are a smart model. I am giving giving you some data regarding an image released by a content creator - (1)
    captions (2) keywords (3) image resolution i.e. (width, height) (4) release date (5) approximate number of downloads
    that the creator wants to achieve (6) approximate number of forwards that the creator wants to achieve (7) approximate
    number of impressions/views that the creator wants to achieve
You have to predict following attributes of the image: (1) exact number of downloads that the image will get (2) exact number
    of forwards that the image will get (3) exact number of impressions/views that the image will get.
Now, predict the attributes for the following image: [ captions: ""Movie slapstick vector illustration . Behind the scenes
     inscription on flapper "", keywords: ""behind the scenes, slapstick, flapper, movie, cinema, scene, logo, frame, film,
     duration, behind, act, black, cameraman, clip, date, director, entertainment, flap, footage, gray, hollywood, icon,
     illustration, inscription, operator, screen, shooting, sign, signal, symbol, television, theater, time, timecode, tv,
     vector, video, view, white "", image resolution: ""(4096, 4096)"", release date: ""2017-06-11"", approximate number of
     downloads that the creator wants to achieve: ""5"", approximate number of forwards that the creator wants to achieve:
     ""31"", approximate number of impressions/views that the creator wants to achieve: ""914"" ] Answer properly in JSON
     format. Do not include any other information in your answer. "

Output: {"exact downloads": 5, "exact forwards": 32, "exact impressions": 864}
```

Listing 5: Engagement Finetuning Verbalization Pattern (4): No engagement in input and asking the model to simulate exact engagement in output

```
Input: "You are a smart model. I am giving giving you some data regarding an image released by a content creator - (1)
    captions (2) keywords (3) image resolution i.e. (width, height) (4) release date
You have to predict following attributes of the image: (1) exact number of downloads that the image will get (2) exact number
    of forwards that the image will get (3) exact number of impressions/views that the image will get
Now, predict the attributes for the following image: [ captions: ""Company employees working in software development and
    designer office "", keywords: ""business, office, meeting, collaegue, successful, workplace, analysis, architect,
    coworker, discussion, entrepreneur, marketing, professional, company, employee, occupation, software, work, worker,
    team, people, brainstorming, cooperation, corporate, project, strategy, teamwork, together, computer, colleagues, young,
    diverse, collaboration, design, developer, group, ideas, management, smiling, multiethnic, place, plan, research,
    startup, technology, women, programmer, architects "", image resolution: ""(4035, 2690)"", release date: ""2020-09-29""
    ]. Answer properly in JSON format. Do not include any other information in your answer."

Output: {"exact downloads": 1, "exact forwards": 1, "exact impressions": 186}
```

# I  TRAIN-TIME OPTIMIZATION ON OTHER TEXT-TO-IMAGE MODELS

We extended our train-time optimization experiments beyond Stable Diffusion 1.4 (Rombach et al., 2022) to include more advanced text-to-image models, Pixart-Sigma (Chen et al., 2024a) and SDXL (512 resolution) (Podell et al., 2023). Specifically, we performed Preferred Finetuning (PFT) on high-engagement images, as described in Section 4.2. The results in Table 6 show strong improvements in image engagement reward as predicted by EngageNet. This demonstrates the effectiveness of the proposed method across multiple text-to-image models.

Table 6: Performance gains on the EngagingImageNet test dataset achieved by applying Preferred Finetuning (PFT) on different text-to-image models, measured by EngageNet's engagement reward.

| Images | Training Config. | High | Medium | Low |
|---|---|---|---|---|
| SD 1.4 | Preferred Finetuning (PFT) on | 56.1489 | 51.6949 | 38.7680 |
| EOIG SD 1.4 | High engagement Images | 62.0390 | 56.1082 | 43.1206 |
| Pixart-Sigma | Preferred Finetuning (PFT) on | 65.8089 | 59.9771 | 46.2870 |
| EOIG Pixart-Sigma | High engagement Images | 68.6384 | 61.9490 | 51.6297 |
| SDXL-512 | Preferred Finetuning (PFT) on | 48.8769 | 44.4127 | 35.1069 |
| EOIG SDXL-512 | High engagement Images | 73.9467 | 64.6240 | 48.3155 |

## J    REPURPOSING ENGAGENET FOR DESIGN SPECIFICATION GENERATION (DSG)

### J.1    TRAINING FOR DESIGN SPECIFICATION PREDICTION TASK

Prior works (Bhattacharya et al., 2023) have demonstrated the capability of language-only pre-trained models like GPT-3 and Vicuna to infer information about visual content without explicit visual reasoning training. Recent models such as BLIP (Li et al., 2023), Llava (Liu et al., 2023), MiniGPT-4 (Zhu et al., 2023), and GPT-4 (OpenAI, 2023) have shown language models' ability to 'see' by incorporating visual branches (often a combination of ViT (Dosovitskiy et al., 2020) and Qformer (Li et al., 2023)) and training them with image-language instructions to answer image-related questions. However, our findings (Table 7) reveal that neither pretraining nor further instruction tuning gives a language model the ability to simulate the downstream engagement of an image-based communication or reason about how a more engaging image should look like. Further, we also find that in-context learning, while successful in many other domains, does not perform well in engagement-related tasks. Therefore, to teach a language model about image content and downstream performance, we further train the Llama LLM.

To teach Llama about an image and its downstream engagement, we perform engagement fine-tuning (Khandelwal et al., 2023). We design four types of engagement-finetuning instructions (Listings 2-5). The idea is to *verbalize* an image using image perception models like color extractor, tones extractor, object, and coordinate detector and convert it to natural language. Then, the image caption, keywords, the required engagement level, date, and marketer information is fed as input to the LLM and asked to output the image verbalization. This way, the LLM learns to map the image prompt and engagement level to image verbalization.

We train the LLM on the train set of EngagingImageNet data. In Listings 2-3, we provide the image caption, keywords, date, and required engagement level as inputs to the model. Our aim is to train the model to predict a design specification comprising colors and tones with their spatial coverage as well as objects with their bounding boxes, that should be reflected in the image. Moreover, we observe improved learning in the language model for engagement-conditioned image generation when introducing a 20% noise in the engagement. We then task the model to rectify this noise in the output, simultaneously generating the verbalization of the engagement-conditioned image in Listings 4-5.

### J.2    RESULTS FOR ENGAGEMENT-CONDITIONED DESIGN SPECIFICATION PREDICTION TASK

To generate engagement-conditioned image verbalization, we compare several models: in-context trained GPT-3.5 and GPT-4, engagement-finetuned Llama (EngageNet), and Llama fine-tuned on image verbalization but without user engagement information. By comparing against a fine-tuned Llama trained on the same instruction as EngageNet, except with the inclusion of engagement tokens, we aim to isolate the impact of engagement tokens on improving generated engagement-conditioned image verbalizations, independent of the instruction tuning process. We assess all models across multiple metrics that evaluate the extent to which the generated verbalizations align with ground truth in terms of colors, tones, objects, and their positions. Intersection over Union (IoU) metrics gauge the overlap between ground truth and generated constructs (colors and objects), while similarity metrics measure cosine similarity between ground truth and generated constructs (colors, objects).

Table 7: Performance of all models on the engagement-optimized design specification generation (DSG) task across different engagement-level images in EngagingImageNet data. It is noteworthy that (i) EngageNet outperforms larger sized GPT-3.5, 4 and also the sized Llama model fine-tuned on the same data (without including engagement tokens). (ii) In-context learning does not work well in the engagement-conditioned design specification generation domain.

| | | | EngagingImageNet Data | | | | | | | | |
| | | | Colours | | | | Tones | Objects | | | |
| Model | Engagement Optimized | Engagement | IOU ↑ | Cosine Similarity ↑ | RGB distance ↓ | Coverage RMSE ↓ | Coverage RMSE ↓ | IOU ↑ | Cosine Similarity ↑ | Normalised Area RMSE ↓ | Relative Position Error ↓ |
|---|---|---|---|---|---|---|---|---|---|---|---|
| Finetuned Llama | No | High | 0.3717 | 0.8725 | 0.2855 | 0.1694 | 0.1957 | 0.2547 | 0.8071 | 0.2612 | 0.3078 |
| | | Low | 0.3362 | 0.8602 | 0.2223 | 0.1811 | 0.2339 | 0.2743 | 0.8047 | 0.2421 | 0.2954 |
| Finetuned Llama (EngageNet) | Engagement Finetuning | High | 0.4065 | 0.8898 | 0.2795 | 0.1507 | 0.1718 | 0.2732 | 0.8122 | 0.2509 | 0.3054 |
| | | Low | 0.4531 | 0.8791 | 0.2084 | 0.1443 | 0.1848 | 0.3455 | 0.8228 | 0.2373 | 0.2889 |
| 3-shot GPT-3.5 | In-context learning | High | 0.214 | 0.7765 | 0.2851 | 0.1773 | 0.396 | 0.1085 | 0.6621 | 0.3090 | 0.3651 |
| | | Low | 0.2175 | 0.7781 | 0.2254 | 0.2118 | 0.3347 | 0.1338 | 0.6749 | 0.3098 | 0.3573 |
| 5-shot GPT-3.5 | In-context learning | High | 0.2137 | 0.7704 | 0.2743 | 0.1976 | 0.324 | 0.1011 | 0.6456 | 0.3160 | 0.3622 |
| | | Low | 0.2191 | 0.7705 | 0.2176 | 0.2449 | 0.3186 | 0.1264 | 0.656 | 0.3150 | 0.3615 |
| 3-shot GPT-4 | In-context learning | High | 0.2421 | 0.7887 | 0.2726 | 0.192 | 0.304 | 0.1035 | 0.6316 | 0.3137 | 0.3666 |
| | | Low | 0.2405 | 0.793 | 0.2332 | 0.2094 | 0.3037 | 0.1419 | 0.6604 | 0.3248 | 0.3763 |
| 5-shot GPT-4 | In-context learning | High | 0.2437 | 0.7905 | 0.2702 | 0.1864 | 0.2937 | 0.1008 | 0.6136 | 0.3111 | 0.3782 |
| | | Low | 0.2448 | 0.7924 | 0.2278 | 0.2144 | 0.2944 | 0.1464 | 0.6406 | 0.3301 | 0.3857 |

Coverage errors determine the how closely the proportion of ground truth and predicted constructs (colors, tones) in the image match. Additionally, we calculate differences in predicted and ground truth areas and locations for objects, accounting for semantically similar objects (such as sofa and couch). Further details on these metrics and their formulas can be found in Appendix J.3.

Table 7 displays the outcomes. The results indicate that engagement fine-tuning enables EngageNet to achieve superior performance across all metrics, surpassing both equivalently sized fine-tuned Llama and 10x larger instruction-tuned GPT-3.5 and GPT-4. Furthermore, in-context learning demonstrates subpar performance, with both the three and five-shot models displaying similar results.

## J.3 EVALUATION METRICS FOR DESIGN SPECIFICATION PREDICTION

- **Colours IOU**: The intersection over union between set $C^G$ of colours in the ground truth image verbalization and set $C^P$ of colours in the predicted image verbalization is computed as:

$$IOU(C^G, C^P) = \frac{|C^G \cap C^P|}{|C^G \cup C^P|} \tag{4}$$

- **Colours similarity**: For the ground truth colour set $C^G = c_1^G, c_2^G, ..., c_i^G$ and predicted colour set $C^P = \{c_1^P, c_2^P, ..., c_j^P\}$, we correspondingly obtain the sets of word vectors $W^G = \{w_1^G, w_2^G, ..., w_i^G\}$ and $W^P = \{w_1^P, w_2^P, ..., w_j^P\}$, using Spacy [†]. For some similarity threshold $\tau$, the mean cosine similarity is computed as follows:

$$\frac{\sum_{i=1}^{|C^G|} \sum_{j=1}^{|C^P|} cos(w_i^G, w_j^P).I(w_i^G, w_j^P, \tau)}{\sum_{i=1}^{|C^G|} \sum_{j=1}^{|C^P|} I(w_i^G, w_j^P, \tau)} \tag{5}$$

where $I(w_i^G, w_j^P, \tau)$ is an indicator function defined as:

$$I(w_i^G, w_j^P, \tau) = \begin{cases} 1 & \text{if } cos(w_i^G, w_j^P) > \tau \\ 0 & \text{otherwise} \end{cases} \tag{6}$$

We take $\tau = 0.7$ in our experiments.

- **Colours RGB distance**: Given the ground truth colour set $C^G = c_1^G, c_2^G, ..., c_i^G$ and predicted colour set $C^P = \{c_1^P, c_2^P, ..., c_j^P\}$, we map each colour to its RGB value to obtain the sets $W^G = \{w_1^G, w_2^G, ..., w_i^G\}$ and $W^P = \{w_1^P, w_2^P, ..., w_j^P\}$ where each element in the sets is a $3 \times 1$ dimensional vector of RGB values. For some distance threshold $\tau$, the mean euclidean distance is calculated as follows:

$$\frac{\sum_{i=1}^{|C^G|} \sum_{j=1}^{|C^P|} distance(w_i^G, w_j^P).\mathbb{I}(w_i^G, w_j^P, \tau)}{\sum_{i=1}^{|C^G|} \sum_{j=1}^{|C^P|} \mathbb{I}(w_i^G, w_j^P, \tau)} \tag{7}$$

[†] https://spacy.io/

where $\mathbb{I}(w_i^G, w_j^P, \tau)$ is an indicator function defined as:

$$\mathbb{I}(w_i^G, w_j^P, \tau) = \begin{cases} 1 & \text{if } distance(w_i^G, w_j^P) < \tau \\ 0 & \text{otherwise} \end{cases} \tag{8}$$

We take $\tau = 0.5$ in our experiments.

- **Colours coverage RMSE**: Consider the intersection $I = C^G \cap C^P$ of ground truth and predicted colour sets. The root mean squared error between the area covered by colours present in both ground truth and predicted image is calculated as follows:

$$RMSE = \sqrt{\frac{1}{|I|} \sum_{i=1}^{|I|} (coverage(c_i^G) - coverage(c_i^P))^2} \tag{9}$$

- **Tones coverage RMSE**: Consider the intersection $I = T^G \cap T^P$ of ground truth and predicted image tones. The root mean squared error between the proportion of tones in ground truth and predicted image is calculated as follows:

$$RMSE = \sqrt{\frac{1}{|I|} \sum_{i=1}^{|I|} (coverage(t_i^G) - coverage(t_i^P))^2} \tag{10}$$

- **Objects IOU**: The intersection over union between set $O^G$ of objects in the ground truth image verbalization and set $O^P$ of objects in the predicted image verbalization is computed as:

$$IOU(O^G, O^P) = \frac{|O^G \cap O^P|}{|O^G \cup O^P|} \tag{11}$$

- **Objects similarity**: For the ground truth set of objects $O^G = \{o_1^G, o_2^G, ..., o_i^G\}$ and set of predicted objects $O^P = \{o_1^P, o_2^P, ..., o_j^P\}$, we correspondingly obtain the sets of word embeddings $W^G = \{w_1^G, w_2^G, ..., w_i^G\}$ and $W^P = \{w_1^P, w_2^P, ..., w_j^P\}$, using Spacy. For some similarity threshold $\tau$, the mean cosine similarity is computed as follows:

$$\frac{\sum_{i=1}^{|O^G|} \sum_{j=1}^{|O^P|} cos(w_i^G, w_j^P).\mathbb{I}(w_i^G, w_j^P, \tau)}{\sum_{i=1}^{|O^G|} \sum_{j=1}^{|O^P|} \mathbb{I}(w_i^G, w_j^P, \tau)} \tag{12}$$

where $\mathbb{I}(w_i^G, w_j^P, \tau)$ is an indicator function defined as:

$$\mathbb{I}(w_i^G, w_j^P, \tau) = \begin{cases} 1 & \text{if } cos(w_i^G, w_j^P) > \tau \\ 0 & \text{otherwise} \end{cases} \tag{13}$$

We take $\tau = 0.7$ in our experiments.

- **Normalised objects area RMSE**: As described above, consider the sets of word vectors of objects present in the ground truth image $O^G = \{o_1^G, o_2^G, ..., o_i^G\}$ and predicted image $O^P = \{o_1^P, o_2^P, ..., o_j^P\}$. Given the ground truth image area $A^G = width \times height$ and a similarity threshold $\tau$, we first compute the mean squared error between the areas of bounding boxes of similar objects in the ground truth and predicted image, weighted by the proportion of each object in the ground truth image and its cosine similarity with the object in the predicted image. Further, we take the square root of the error thus obtained and normalise it by $A^G$ to achieve the desired metric, as follows:

$$MSE = \frac{\sum_{i=1}^{|O^G|} \sum_{j=1}^{|O^P|} \{(area(o_i^G) - area(o_j^P))^2 . \frac{area(o_i^G)}{A^G} . \frac{1}{cos(w_i^G, w_j^P)}\} . \mathbb{I}(w_i^G, w_j^P, \tau)}{\sum_{i=1}^{|O^G|} \sum_{j=1}^{|O^P|} \mathbb{I}(w_i^G, w_j^P, \tau)} \tag{14}$$

$$Normalised\ RMSE = \frac{\sqrt{MSE}}{A^G} \tag{15}$$

where $\mathbb{I}(w_i^G, w_j^P, \tau)$ is an indicator function as described above. We take $\tau = 0.7$ in our experiments.

- **Normalised relative position error**: Following a similar approach as explained above, we compute the mean euclidean distance between the centroids of bounding boxes of similar objects weighted by the cosine similarity of objects present in the ground truth and predicted images and normalise it by the length of diagonal in the ground truth image $D^G$:

$$RPE = \frac{\sum_{i=1}^{|O^G|} \sum_{j=1}^{|O^P|} \{(distance(centroid(o_i^G), centroid(o_j^P) \cdot \frac{1}{cos(w_i^G, w_j^P)}\} . \mathbb{I}(w_i^G, w_j^P, \tau)}{\sum_{i=1}^{|O^G|} \sum_{j=1}^{|O^P|} \mathbb{I}(w_i^G, w_j^P, \tau)} \tag{16}$$

$$Normalised\ RPE = \frac{RPE}{D^G} \tag{17}$$

where $\mathbb{I}(w_i^G, w_j^P, \tau)$ is the aforementioned indicator function. As before, we take $\tau = 0.7$ in our experiments.

## K  PERFORMANCE ALIGNMENT OF STABLE DIFFUSION USING DESIGN SPECIFICATION GENERATION (DSG) REWARD

### K.1  DDPO ADDITIONAL DETAILS

The denoising process in diffusion models is a multi-step recursive process with a pre-specified finite number of steps. In DDPO Black et al. (2023), this denoising process is viewed as a finite horizon Markov decision process (MDP), where the state comprises of the current context, number of steps left in the process and the current denoised image. The action to be taken is to predict the next image using this state.

The image forming the initial state is sampled from a standard normal distribution. Mathematically, a finite horizon MDP is defined as a tuple $\{T, \mathcal{S}, \mathcal{A}, P, R\}$, where these components are defined as:

1. $T$ is the horizon or the number of steps in the MDP
2. $\mathcal{S}$ is the state space. Here it comprises of three components, the context $c$, the current number of steps left in the denoising process, $t$, and the current denoised image representation (a given vector encoding of the image), $x_t$. The initial or starting state has the context $c_0$ given as input, the number of steps left at the beginning, $t_0 = T$ and the initial image representation is sampled from a normal distribution of appropriate dimension, $x_0 \sim \mathcal{N}(0, I)$.
3. $\mathcal{A}$ is the action space, and here it is the space comprising of all image representations $x$. If $x$ is a $d-$dimensional vector, then $\mathcal{A} = \mathbb{R}^d$.
4. $P : \mathcal{S} \times \mathcal{A} \to \Delta(\mathcal{S})$ is the transition function. Here, we specify $P$ separately for each of the three components of the state as $P_c = \delta(c_t)$, $P_t = \delta(t - 1)$, and $P_x = \delta(a_t)$, where the current state is $c_t, t, x_t$, current action $a_t = x_{t-1}$, and $\delta(\cdot)$ is the Dirac delta distribution.
5. $R : \mathcal{S} \times \mathcal{A} \to \mathbb{R}$ is the reward function that takes a state and action as input and returns a scalar reward. We generate this scalar reward signal using EngageNet.

As the agent acts in the MDP, it produces trajectories, which are sequences of states and actions: $\tau = (s_0, a_0, s_1, a_1, \ldots, s_T, a_T)$. The reinforcement learning (RL) objective is for the agent to maximize the expected cumulative reward over trajectories sampled from its policy. Assuming a fixed sampler, the diffusion model generates a sample distribution $p_\theta(x_0 \mid c)$. The denoising diffusion RL objective is to maximize a reward signal $r$ which is defined based on the generated samples and their corresponding contexts:

$$J_{\text{DDRL}}(\theta) = \mathbb{E}_{c \sim p(c), x_0 \sim p_\theta(x_0|c)} [r(x_0, c)]$$

for some context distribution $p(c)$.

### K.2  DESIGN SPECIFICATION GENERATION (DSG) REWARD

The steps of constructing the reward function based on design specification generation are given below 14b:

- We featurize the image generated by stable diffusion to obtain features (including colors, tones, objects, and their positions) that EngageNet is meant to predict as part of a design

specification conditioned on contextual information such as marketer, expected likes, tweet content, image caption, *etc.*

- Based on the above conditioning factors, we use EngageNet to predict the logits of the verbalized features of the image generated by stable diffusion as described in the previous step.
- We now have one logit per text token as EngageNet's output. To convert this to a scalar score, we compute the probabilities of each token and then add them.

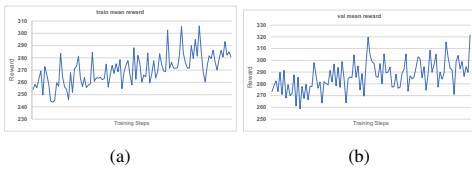

(a)                              (b)

Figure 7: Reward curves for the performance alignment of stable diffusion on EngagingImageNet (train (a) and validation (b) sets)

Table 8: Results for the performance of various models on the EngagingImageNet for the engagement-optimized image generation task. Results are computed on the EngageNet Design Specification Generation (DSG) reward (§K) as well as other metrics reported in the literature.

| Images | Engagement | Reward ↑ | Other Metrics | | |
|---|---|---|---|---|---|
| | | | FID ↓ | Aesthetic Score ↑ | CLIP Score ↑ |
| Base Stable Diffusion | High | 242.545 | 34.958 | 5.221 | 33.346 |
| | Low | 238.471 | 42.999 | 4.925 | 31.705 |
| Fine-tuned on High Engagement Images | High | 239.023 | 26.023 | 4.850 | 32.210 |
| Stable Diffusion | Low | 223.619 | 37.497 | 4.433 | 30.979 |
| EngageNet aligned | High | 254.918 | 36.546 | 5.341 | 33.379 |
| Stable Diffusion (EOIG-SD) | Low | 247.597 | 49.492 | 5.087 | 31.719 |

## L  BROADER IMPACTS AND LIMITATIONS

Our work on assessing and improving the engagement of text-to-image generation models introduces several societal considerations that require careful examination. We aim to provide a comprehensive analysis of the potential impacts, highlighting both contributions to the field and the precautions necessary for responsible development and deployment of engagement-optimizing technologies.

1. The ability of models to enhance user engagement raises important societal concerns around responsible deployment and potential misuse. Quantifying these risks is essential for developing appropriate safeguards. However, studying user engagement, particularly in uncontrolled environments, poses ethical challenges. For instance, investigating engagement through manipulated or highly optimized content could influence user behavior in unintended or harmful ways. To mitigate this risk, we have limited our research to controlled environments and observational analyses, ensuring that experimental insights are gathered without causing real-world harm.

2. To foster responsible use of our research and datasets, we will release an Acceptable Use Policy explicitly prohibiting the misuse of our dataset for generating content aimed at harmful or deceptive purposes. This includes banning its use in abusive contexts (e.g., creating deceptive ads or manipulative imagery) and sensitive applications such as political propaganda. We will actively monitor compliance with this policy and encourage others in the research community to adopt similar ethical guidelines when using engagement-optimizing models. Importantly, our dataset is PII-free, ensuring that no personal information of individuals is included. Our dataset compilation adheres to Twitter's API terms of service. We used the Twitter API from 2015-2023 for data collection, and our dataset release will comply with all restrictions outlined in Twitter's Developer Agreement and Policy, available at `https://developer.x.com/en/developer-terms/agreement-and-policy`.

3. We plan to release the dataset and evaluation frameworks in stages, starting with the release of our benchmark and engagement arena. This staged release will help familiarize the research community with the methodologies we use to assess engagement in generated content. By gradually releasing the dataset (in batches of 20%), we will closely monitor

how models perform in enhancing engagement. Initially, the dataset will only be available in a controlled environment, enabling us to manage usage and address emerging concerns. Throughout this process, we will actively engage with the research community, encouraging responsible use and urging fellow researchers to contribute additional persuasion-related data using our infrastructure. We also encourage the research community to contribute additional data to expand our evaluation framework. This approach balances the need for research progress with ethical responsibility and community involvement.

4. We recognize the dual-use potential of models designed to optimize engagement. While this technology can be beneficial in fields like education or user-centered design, it also poses risks of misuse in deceptive contexts. Drawing parallels to ethical discussions on persuasive technologies, we believe that transparency and safeguards in dataset design can mitigate the potential for harm. The insights gained from understanding user engagement can aid in the responsible development of future AI systems.

5. PII Removal and Data Collection: To protect user privacy, we have implemented measures to remove all personally identifiable information (PII). Our dataset is compiled without collecting sensitive personal data, focusing solely on public, non-individualized information. All references to specific users or personal identifiers have been removed. Additionally, we collect only aggregate metrics (e.g., overall user interaction data) to measure engagement trends without compromising individual privacy.

6. In this work, we specifically focus on the engagement optimization capabilities of text-to-image generation models. We introduce benchmarks and evaluation methodologies for measuring user engagement with AI-generated images and develop techniques to enhance this engagement. Our findings suggest that engagement with generated content can be improved not just by increasing model size but also through targeted training strategies. Furthermore, engagement patterns observed in one domain (e.g., Twitter) often transfer to other domains (e.g., Pinterest), which broadens the applicability of our findings.

### L.1 LIMITATIONS

In this paper, we examine a single aspect of engagement. In real-world applications, user engagement often occurs in sequential or multi-stage interactions, which we plan to address in future research. Additionally, this work is focused on English-language data; we aim to extend our findings to other languages in subsequent studies. Furthermore, the impact of audience dependence on engagement has not been studied extensively in this paper, partly due to the absence of publicly available datasets. We plan to work on collecting such datasets to explore this effect in future work. In the current work, we do not explicitly account for potential unobserved confounders, such as external events and trending topics, which may influence engagement. These limitations underscore areas for further research and caution against over-generalizing our findings to more complex real-world scenarios.

## M ADDITIONAL FIGURES

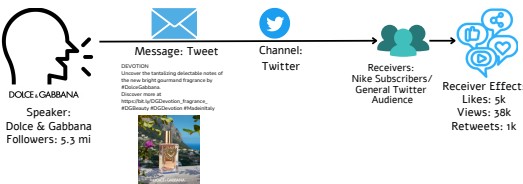

Figure 8: Any message is created to serve an end goal. For marketers, the end goal is to bring in the desired receiver effect (behavior) (like clicks, purchases, likes, and customer retention). The figure presents the key elements in the communication pipeline - the marketer, message, channel, receivers, and finally, the receiver effect. Traditionally, image generation is optimized on metrics such as aesthetics and FID. For effective communication, the image generation process needs to be optimized on the receiver effect (other than the traditional metrics).

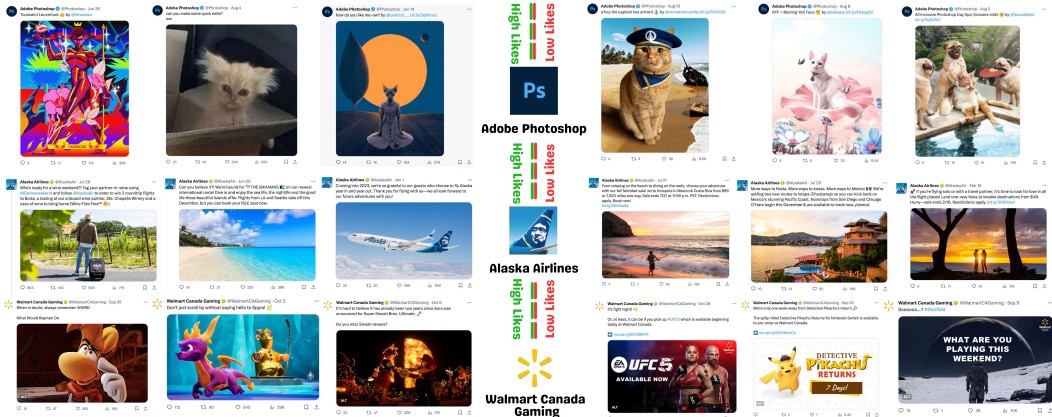

Figure 9: Sample media and tweets from enterprise accounts in the EngagingImageNet dataset. It can be noted, for example, in the Adobe Photoshop tweets, that the media does not differ significantly in aesthetics or objects themselves (all of them are cats). Despite that, there is much difference in the image likes, indicating that viewer engagement is distinct from other optimization objectives such as aesthetics or prompt adherence.

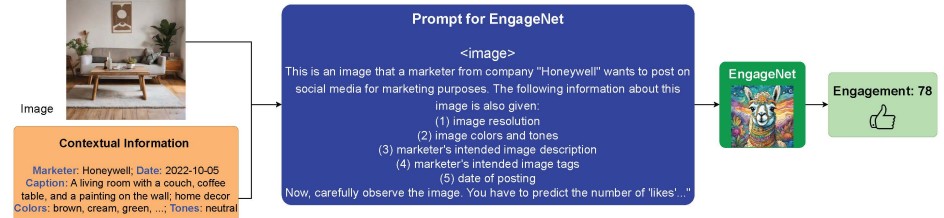

Figure 10: Visual Instruction Finetuning of EngageNet on EngagingImageNet dataset. The EngageNet model is trained to predict the engagement score of an image on a 0-100 scale, conditioned on marketer provided metadata comprising the company, image resolution, image colours and tones with their spatial coverage, marketer's intended image description and tags, and the date of releasing the image on social media.

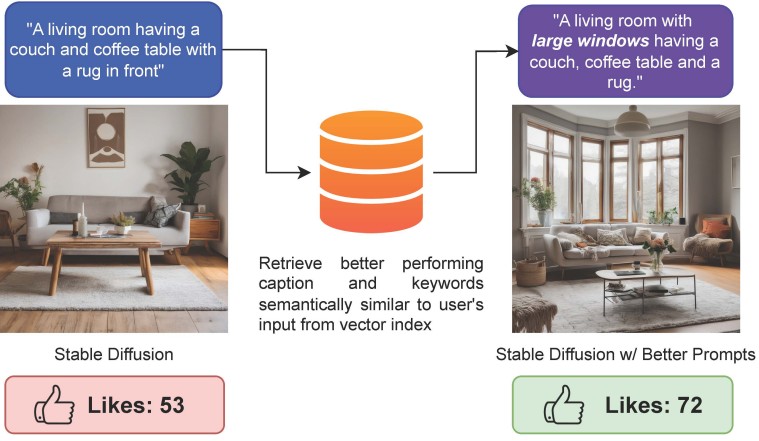

Figure 11: Retrieval framework for conditioning text-to-image models on higher engagement prompts as described in Section 4.1. The retrieved prompts may incorporate image characteristics that have been empirically shown to improve image engagement.

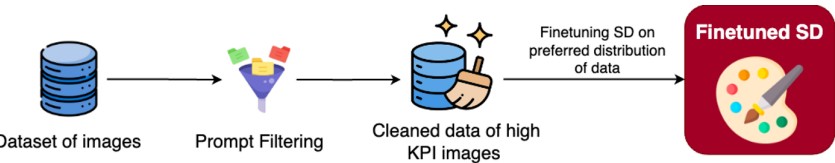

Figure 12: Illustration depicting supervised finetuning of stable diffusion model on high-liked images from EngagingImageNet dataset as described in Section 4.2. This method of finetuning U-Net module on preferred data distribution results in generating more engaging images.

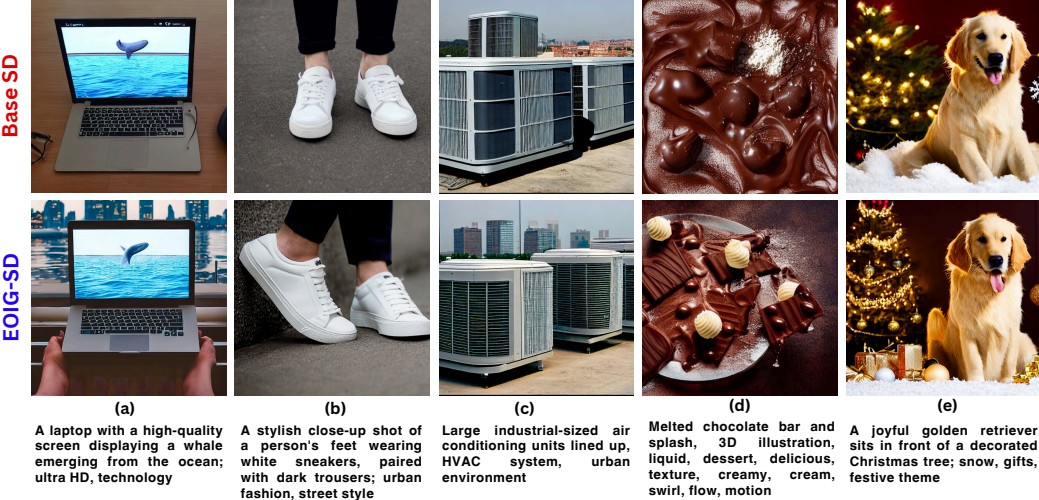

Figure 13: Comparison of generated images - EOIG-SD *vs* Base stable diffusion. Engagement optimization helps the model to learn to generate persuasion skills. EOIG-SD generates better product photography (a,c,d), model photography (b), generates images with social appeal and social identity (a,c), and learns temporal patterns (e) (prominently Christmas-themed image of dog)

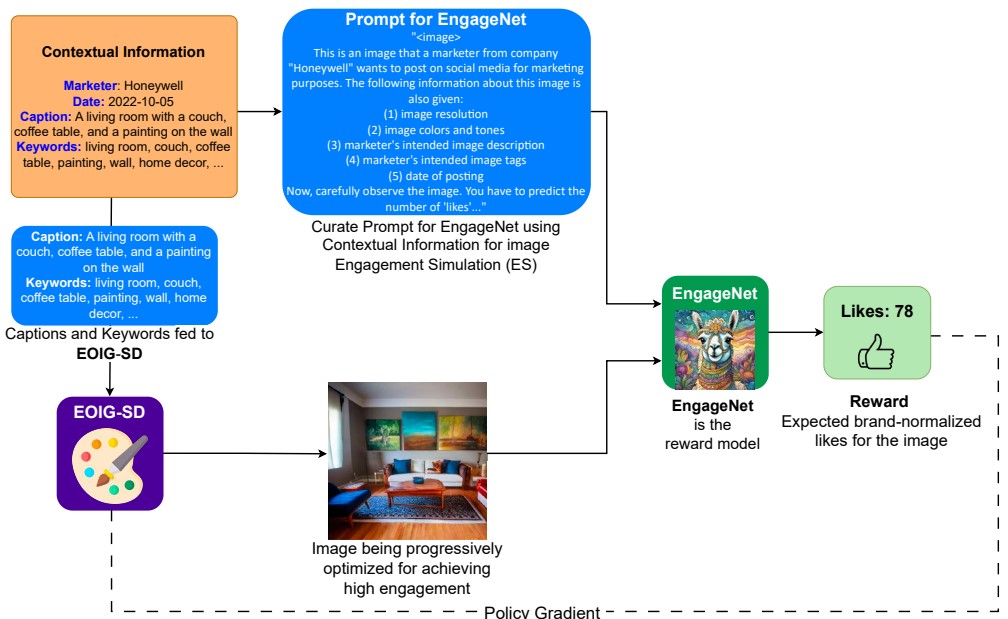

(a) Architecture of the proposed pipeline for training stable diffusion for the objective of engagement-optimised image generation (EOIG) using Engagement Simulation (ES) reward function as described in Section 4.3. EngageNet predicts the engagement level of images generated by stable diffusion. The scalar rewards are used to guide stable diffusion to produce progressively higher engagement images. The resulting diffusion model is called EOIG-SD (RLHF-ES).

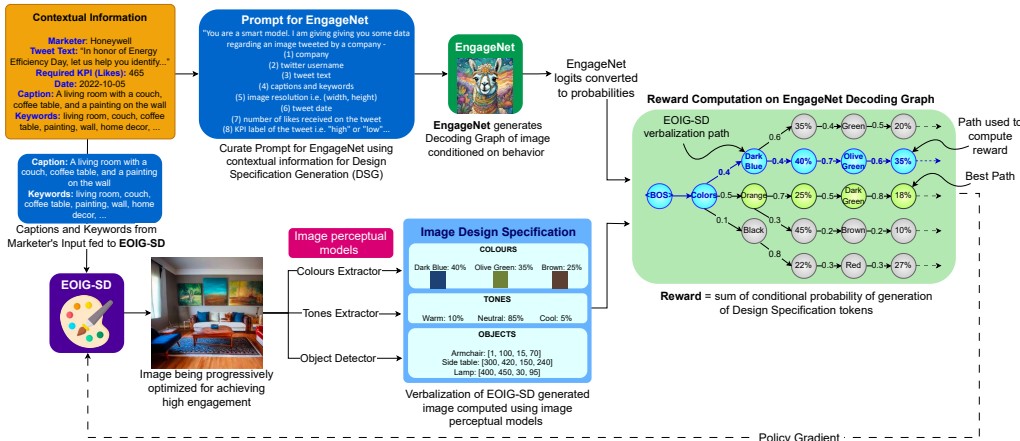

(b) Architecture of the proposed pipeline for training stable diffusion for the objective of engagement-optimised image generation (EOIG) using Design Specification Generation (DSG) reward function as described in Section 4.3. EngageNet trained in the manner described in Appendix J possesses the capability to generate verbal descriptions comprising colors, tones, objects and their locations of an image based on conditioning factors such as the company, time, image caption and viewer likes. Thus, EngageNet inherently understands the design details of an image, for a given engagement level and caption. We leverage this EngageNet as a reward model to train stable diffusion such that the images generated by it have a design specification aligned with those of higher engagement image. EOIG-SD takes a prompt and generates an image, which then undergoes verbalization via image perception models. Its objective is to create images that, when verbalized, closely resemble the engagement-conditioned verbalization generated by EngageNet. The verbalized output of EOIG-SD is fed into the reward model. We ask EngageNet to predict the logits for this image verbalization, using which a reward is computed for EOIG-SD, indicating how closely this verbalized output aligns with EngageNet. This reward value serves as feedback for EOIG-SD in the form of policy gradient, aiding in its continual improvement and refinement within the image generation process. Thus, this pipeline trains EOIG-SD to generate engagement-optimized images by gradually aligning its output with EngageNet.

Figure 14: Aligning Stable Diffusion for higher engagement using DDPO algorithm (Black et al., 2023) using two types of reward functions - (a) Engagement Simulation (ES) and (b) Design Specification Generation (DSG)

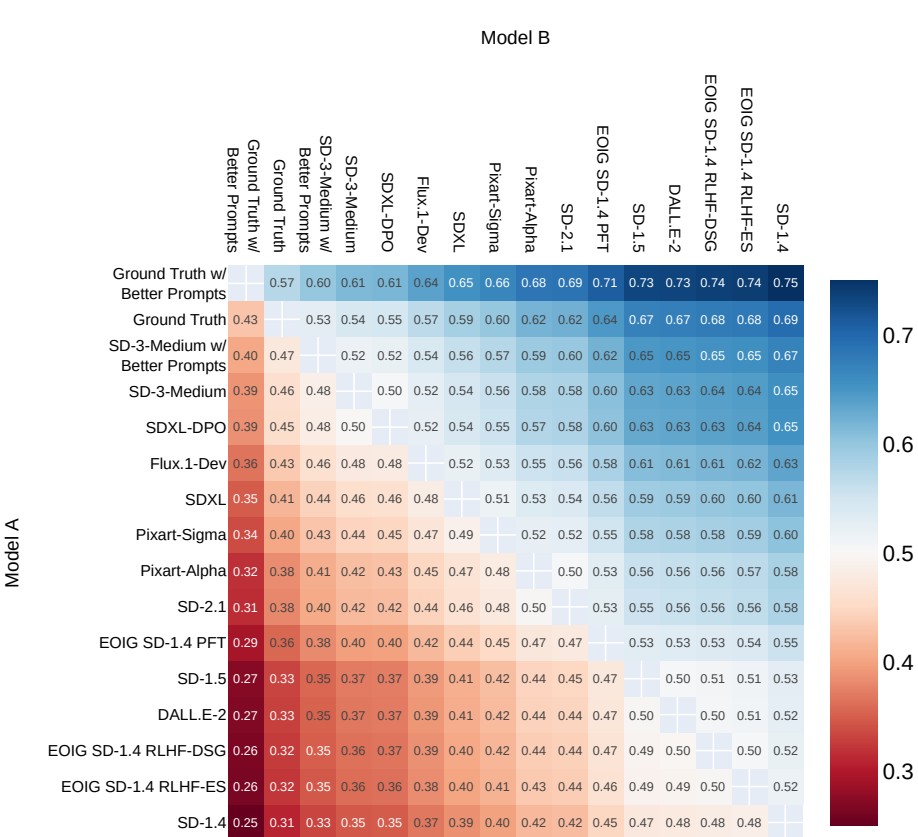

Figure 15: Win Rates of different models against each other in the Image Engagement Arena

