# OpenReview forum: "Measuring And Improving Engagement of Text-to-Image Generation Models"
_ICLR.cc/2025/Conference — ICLR 2025 Poster_

### Official Review · Reviewer_wzeb · 2024-11-03

**Soundness:** 3
**Presentation:** 3
**Contribution:** 3
**Rating:** 6
**Confidence:** 4

**Summary:**

This paper introduces a novel consideration in the text-to-image generation task: human engagement. The authors first collect a large dataset, EngagingImageNet, from an online platform. They then train EngageNet to assess the engagement level of generated images based on auxiliary information embedded in the prompts. Additionally, they propose methods to leverage EngageNet to enhance the performance of text-to-image models. Finally, the authors introduce Engagement Arena to assess the engagement levels of different text-to-image models. This work presents a comprehensive framework for evaluating and improving engagement of text-to-image models.

**Strengths:**

1. The collected dataset EngagingImageNet is a strong foundation, as authors provide the detailed introduction of obtaining and filtering this datasets. This dataset encompasses multimodal information for each image, supporting not only engagement assessment but also various other applications.
2. The author trained the EngageNet with several tricks, obtaining a superior performance. The release of such model would offer both academia and industry a new perspective for evaluating generated images.
3. This paper introduces multiple strategies for enhancing the engagement of generated images, during run-time and train-time. They achieve notable improvement using these approaches.

**Weaknesses:**

1. The paper would benefit from a more detailed explanation of the DDPO method utilized, including the hyper-parameters and loss terms.

2. The paper lacks a user study or online experiment to evaluate the engagement of images generated by the fine-tuned models. It is unclear whether the fine-tuned models genuinely improve engagement or merely optimize to satisfy the reward model.

3. The paper lacks discussion of some relevant works:

[1] Rich Human Feedback for Text-to-Image Generation, CVPR 2024.
[2] Towards Reliable Advertising Image Generation Using Human Feedback, ECCV 2024.
[3] Directly Fine-Tuning Diffusion Models on Differentiable Rewards, ICLR 2024.

Moreover, other works on using human feedback to improve image generation models should be considered.

**Questions:**

1. Why did the authors not consider including retweets as part of the engagement indicator?

2. Why was SD 1.4 chosen for train-time optimization, as it is one of the weakest models?

3. Does KPI refer to the normalized likes of an image? So the regression value of KPI is just used in training phase?

4. I would like to see an analysis of the complexity or latency of the proposed components, such as the Retrieval module, since an excessively long generation process would be impractical for real-world applications.

5. All concerns in Weaknesses section.

---

> ### Author Response · Authors · 2024-11-28
> **Author Response to Reviewer wzeb (1/5)**
>
> We sincerely thank the reviewer for the insightful review and feedback. We try to address the reviewer's questions and comments in detail below:
>
>
> ---
> > The paper lacks discussion of some relevant works, other works on using human feedback to improve image generation models should be considered
> ---
> We have tried to discuss works related to human feedback in the current draft (we cover that again below). Further, as suggested by the reviewer, we will expand the discussion of works related to utilising human feedback to train image generation models in the next paper draft. We discuss the different works below:
>
> To align image generation with human preferences, reinforcement learning with human feedback (RLHF) has been utilised for text-to-image generation models [1, 2]. In these approaches, the latent image generation part of the diffusion model (either UNet or a Transformer) is trained using a reward model in the case of DDPO or using user preferences directly in the case of DPO. Both these approaches involve collecting human preference datasets.
>
> Some more recent works are discussed below (as suggested by the reviewer):
>
> Liang et al. [3] introduce a dataset RichHF-18K and a multimodal transformer model (RAHF) that aims to predict detailed human feedback, such as identifying implausible image regions and misaligned text. They collected rich human feedback over 18,000 images from various users. Then they tune the RAHF model to predict implausibility scores, heatmaps of artifact locations, and text-image misalignment. The predicted feedback is utilized to enhance image generation quality through targeted finetuning.
>
> Du et al. [4] curate a large-scale dataset consisting of over one million generated advertising images, each annotated with human feedback regarding their suitability for advertising purposes. This dataset is created using a diverse selection of products from JD.com, featuring generated advertising images alongside corresponding product images with transparent backgrounds and meticulously crafted prompts by professional designers. Additionally, the paper introduces a multimodal model trained on this dataset to simulate human feedback of advertising suitability and automate image evaluation. This model utilizes an efficient recurrent generation process to produce high-quality advertisement images.
>
> Clark et al. [5] propose a method for gradient-based reward finetuning based on differentiation in the diffusion sampling process. This approach, Direct Reward Fine-Tuning (DRaFT) allows for optimising diffusion models by incorporating differentiable rewards based on human feedback. They demonstrate the application of this method on a variety of reward functions, such as aesthetic score [6], PickScore [7] and Human Preference Score [8].
>
> Liang et al. [3], Du et al. [4], Clark et al. [5]  use simulated human feedback to refine the image generation process for specific image quality dimensions of realism [3,4,5], aesthetics [4,5], visual appeal [5] or advertising suitability [4].
>
> Further, several human preference datasets for text-to-image generation have been collected in literature. These include Pick-a-Pic dataset [7], dataset generated from the Stable Foundation Discord channel [8], ImageReward dataset [9] etc. The human preferences in these datasets have been collected by explicitly asking humans to state their choices. These datasets are often accompanied with their own metrics for human preference alignment such as PickScore [7], Human Preference Score [8], ImageReward score [9] etc. While the authors of these research works have shown that these metrics are better aligned with human preferences when compared with conventional metrics such as CLIP score, BLIP score, Aesthetic Score, etc, however, they still need not align with viewer engagement of images. Hence, in our work, we use implicit data about human preferences derived from engagement metrics such as likes. Thus, we explore holistic and user-centric metrics of engagement that measure dimensions beyond traditional image quality. Our contribution is unique in the sense that we prioritize viewer engagement as the core feedback, allowing for engagement-optimized image generation that directly aligns with how users interact with visual content online.
> While traditional metrics like aesthetics, FID, ImageReward, image realism help in making the images look real, aesthetic and structurally correct, optimizing and measuring viewer engagement of images has potential applications in personalization, efficient A/B testing, recommender systems, and even public awareness and information campaigns.
>
> References (continued)

---

> > ### Author Response · Authors · 2024-11-28
> > **Author Response to Reviewer wzeb (2/5)**
> >
> > References:
> > [1] Black, K., Janner, M., Du, Y., Kostrikov, I., & Levine, S. Training Diffusion Models with Reinforcement Learning. In The Twelfth International Conference on Learning Representations.
> > [2] Wallace, B., Dang, M., Rafailov, R., Zhou, L., Lou, A., Purushwalkam, S., ... & Naik, N. (2024). Diffusion model alignment using direct preference optimization. In Proceedings of the IEEE/CVF Conference on Computer Vision and Pattern Recognition (pp. 8228-8238).
> > [3] Liang, Y., He, J., Li, G., Li, P., Klimovskiy, A., Carolan, N., ... & Navalpakkam, V. (2024). Rich human feedback for text-to-image generation. In Proceedings of the IEEE/CVF Conference on Computer Vision and Pattern Recognition (pp. 19401-19411).
> > [4] Du, Z., Feng, W., Wang, H., Li, Y., Wang, J., Li, J., ... & Shao, J. (2024). Towards Reliable Advertising Image Generation Using Human Feedback. In European Conference on Computer Vision.
> > [5] Clark, K., Vicol, P., Swersky, K., & Fleet, D. J. Directly Fine-Tuning Diffusion Models on Differentiable Rewards. In The Twelfth International Conference on Learning Representations.
> > [6] Schuhmann, C., Beaumont, R., Vencu, R., Gordon, C., Wightman, R., Cherti, M., ... & Jitsev, J. (2022). Laion-5b: An open large-scale dataset for training next generation image-text models. Advances in Neural Information Processing Systems, 35, 25278-25294.
> > [7] Kirstain, Y., Polyak, A., Singer, U., Matiana, S., Penna, J., & Levy, O. (2023). Pick-a-pic: An open dataset of user preferences for text-to-image generation. Advances in Neural Information Processing Systems, 36, 36652-36663.
> > [8] Wu, X., Sun, K., Zhu, F., Zhao, R., & Li, H. (2023). Human preference score: Better aligning text-to-image models with human preference. In Proceedings of the IEEE/CVF International Conference on Computer Vision (pp. 2096-2105).
> > [9] Xu, J., Liu, X., Wu, Y., Tong, Y., Li, Q., Ding, M., ... & Dong, Y. (2024). Imagereward: Learning and evaluating human preferences for text-to-image generation. Advances in Neural Information Processing Systems, 36.

---

> ### Author Response · Authors · 2024-11-28
> **Author Response to Reviewer wzeb (3/5)**
>
> ---
> > The paper would benefit from a more detailed explanation of the DDPO method utilized, including the hyper-parameters and loss terms
> ---
>
> Here, we provide details of the DDPO method adopted for aligning Stable Diffusion with engagement. Some of these are discussed are discussed in Section 4.3 and Appendix G.1 of the paper. We will expand and highlight the details better in the next draft version.
>
> The denoising process in diffusion models is a multi-step recursive process with a pre-specified finite number of steps. In DDPO [1], this denoising process is viewed as a finite horizon Markov Decision Process (MDP), where the state comprises the current context, the number of steps left in the process, and the current denoised image. The action to be taken is to predict the next image using this state.
> The image forming the initial state is sampled from a standard normal distribution. Mathematically, a finite horizon MDP is defined as a tuple $\{T, S, A, P, R\}$, where these components are defined as:
>
> 1. $T$ is the horizon or the number of steps in the MDP.
> 2. $S$ is the state space. Here it comprises three components: the context $c$ (i.e., the text prompt used for conditioning the diffusion model), the current number of steps left in the denoising process $t$, and the current denoised image representation (a given vector encoding of the image) $x_t$.
>    The initial or starting state has the context $c_0$ given as input, the number of steps left at the beginning $t_0 = T$, and the initial image representation is sampled from a normal distribution of appropriate dimension, $x_0 \sim \mathcal{N}(0, I)$.
> 3. $A$ is the action space, and here it is the space comprising all image representations $x$. If $x$ is a $d$-dimensional vector, then $A = \mathbb{R}^d$.
> 4. $P : S \times A \rightarrow \Delta(S)$ is the transition function. Here, we specify $P$ separately for each of the three components of the state as:   $\[
>    P_c = \delta(c_t), \quad P_t = \delta(t - 1), \quad P_x = \delta(a_t)
>    \]$ , where the current state is $(c_t, t, x_t)$, the current action $a_t = x_{t-1}$, and $\delta(\cdot)$ is the Dirac delta distribution.
> 5. $R : S \times A \rightarrow \mathbb{R}$ is the reward function that takes a state and action as input and returns a scalar reward.
>    We generate this scalar reward signal using EngageNet. EngageNet is used to estimate the user engagement of images generated by Stable Diffusion. The reward signal is then leveraged to guide Stable Diffusion to generate higher-engagement images, as illustrated in Figure 13a.
>
> As the agent acts in the MDP, it produces trajectories, which are sequences of states and actions:
> $\tau = (s_0, a_0, s_1, a_1, \dots, s_T, a_T). $
> The reinforcement learning (RL) objective is for the agent to maximize the expected cumulative reward over trajectories sampled from its policy. Assuming a fixed sampler, the diffusion model generates a sample distribution $p_\theta(x_0 \mid c)$. The denoising diffusion RL objective is to maximize a reward signal $r$ which is defined based on the generated samples and their corresponding contexts:
>
> $ J_\text{DDRL}(\theta) = \mathbb{E}_{c \sim p(c), \, x_0 \sim p\theta(x_0 \mid c)} \big[r(x_0, c)\big] $
>
> for some context distribution $p(c)$.
>
> As suggested by the reviewer, we will include additional details about the hyperparameters and loss terms of the DDPO method in the next draft of the paper.
>
> **Hyperparameters**:
> - batch size = 8
> - epochs = 100
> - number of samples per epoch = 64
> - learning rate = 5e-6
> - weight decay = 1e-4
> - clip ratio = 1e-4
> - optimizer = AdamW
>
> References:
> [1] Black, K., Janner, M., Du, Y., Kostrikov, I., & Levine, S. Training Diffusion Models with Reinforcement Learning. In The Twelfth International Conference on Learning Representations.
>
>
> ---
> > lacks a user study or online experiment
> ---
> We have initiated the effort to conduct a user study. We have started the process of Ethics Review Board approval. We aim to incorporate the results of the user study by the camera-ready version deadline.

---

> > ### Author Response · Authors · 2024-11-28
> > **Author Response to Reviewer wzeb (4/5)**
> >
> > ---
> > > whether the fine-tuned models genuinely improve engagement or merely optimize to satisfy the reward model.
> > ---
> >
> > We explored the following methods for engagement-driven image generation:
> >
> > - Run-time Optimization:
> >
> >   - Conditioning Stable Diffusion on improved prompts to enhance the quality and engagement potential of generated images.
> >
> > - Train-Time Optimizations:
> >   - Preferred Fine-Tuning (EOIG-PFT): Fine-tuning Stable Diffusion on high-engagement images to learn patterns associated with higher engagement.
> >   - Reinforcement Learning (EOIG-RLHF): Aligning Stable Diffusion with EngageNet-based rewards using DDPO, enabling direct optimization of the reward model.
> >
> > It is important to note that only the EOIG-RLHF suite of models was explicitly trained to optimize directly for the EngageNet-based rewards. The other models, such as those developed through preferred fine-tuning (EOIG-PFT) or prompt conditioning, were not directly trained to maximize EngageNet scores. Despite this, significant increases in the engagement of their generated images were observed shown in Table 4 of the paper.
> >
> > Extensive experiments and comparisons across different training methods and configurations demonstrate that the improvements are not incidental but result from genuine improvements in alignment with engagement objectives. Results from the Engagement Arena benchmarking further validate that fine-tuning leads to models capable of generating images with higher engagement scores, as judged by our evaluation framework.
> >
> > Another point to consider is that the image engagement metric used for evaluating text-to-image models is grounded in real-world data. The evaluation model, EngageNet
> > evaluation model, EngageNet is an engagement-aware VLM was trained using real-world engagement data from the EngagingImageNet dataset. Its predictions have been validated to correlate strongly with actual user engagement metrics (account-normalised likes). This ensures that EngageNet’s predicted engagement scores are aligned with real user engagement.
> >
> > ---
> > > Why did the authors not consider including retweets as part of the engagement indicator?
> > ---
> > Thank you for your insightful question regarding the inclusion of retweets as part of the engagement metric. We acknowledge that retweets can also be an important indicator of content engagement. However, due to operational constraints and limitations in the Twitter API v1, we were unable to collect retweet data for our analysis. As a result, we chose to use likes as the primary engagement metric in our study.

---

> ### Author Response · Authors · 2024-11-28
> **Author Response to Reviewer wzeb (5/5)**
>
> ---
> > Why was SD 1.4 chosen for train-time optimization, as it is one of the weakest models?
> ---
> Thank you for suggesting this experiment.
> In the paper, we performed train-time optimisations on SD 1.4  due to resource constraints. SD 1.4 being of relatively smaller size enabled us to conduct thorough experiments in various training configurations.
> As kindly suggested by the reviewer, during the rebuttal phase, we conducted experiments to apply train-time optimisation over more recent and strong text-to-image models, like Pixart-Sigma and SDXL (512 resolution). Specifically, we performed preferred finetuning on high engagement images (Section 4.2). Results reveal the impact of improvement in image engagement after training.
>
> | Images                    | High     | Medium   | Low      |
> |---------------------------|----------|----------|----------|
> | **Base Model: Pixart Sigma** |          |          |          |
> | Pixart-Sigma               | 65.8089  | 59.9771  | 46.2870  |
> | EOIG-PFT Pixart-Sigma      | 68.6384  | 61.9490  | 51.6297  |
> | **Base Model: SDXL-512**    |          |          |          |
> | SDXL-512                   | 48.8769  | 44.4127  | 35.1069  |
> | EOIG SDXL-512              | 73.9467  | 64.6240  | 48.3155  |
>
> .
> ---
> > Does KPI refer to the normalized likes of an image? So the regression value of KPI is just used in training phase?
> ---
> Yes, KPI refers to the account-normalised likes of the image which reflects its engagement level. It serves as ground truth for training EngageNet for the image engagement prediction (regression) task. EngageNet’s predicted scores serve as a metric to judge the engagement level of images generated by various text-to-image models in the Engagement Arena. For maintaining uniformity and improving clarity, we would rename KPI to engagement in the updated draft of the paper.
>
> ---
> > latency of the proposed components, such as the Retrieval module
> ---
> For the retrieval module, we use FAISS framework to index the DistilBERT vector embeddings of captions belonging to images in the high engagement data subset of the EngagingImageNet train data. Next, for every image caption in the low performance subset of the test data, we retrieve the semantically most similar caption from the corpus of high-performing images.
>
> Once the vector index is created, average retrieval time per query = 0.1852 seconds / query (Averaged across ~2K queries).
> Memory consumption: 1.1 GB GPU RAM; Hardware: Nvidia A100-SXM4

---

> > ### Comment · Reviewer_wzeb · 2024-12-02
> > **Reply to Author Response**
> >
> > Thanks for the authors' detailed response and additional experiments. The authors mostly answered my questions and addressed my concerns. I think it is a solid work and will have a positive impact on the community. However, there is some missing content in the paper, and it seems that the authors did not have time to modify the original paper in this phase. Thus, I hope the authors could include these missing parts, such as the user study, to make this work complete in the camera-ready version. I will keep my score.

---

### Official Review · Reviewer_k6Wd · 2024-11-03

**Soundness:** 3
**Presentation:** 3
**Contribution:** 3
**Rating:** 8
**Confidence:** 3

**Summary:**

This study constructs a large-scale dataset of images paired with user engagement metrics to investigate image engagement and utility in real-world marketing contexts. An engagement-aware visual language model (VLM) is developed to predict viewer engagement. Contemporary techniques are explored to enhance engagement in text-to-image models, including conditioning image generation on optimized prompts, supervised fine-tuning of Stable Diffusion on high-performing images, and reinforcement learning to align Stable Diffusion outputs. The authors also introduce a benchmarking method for text-to-image engagement models, offering the research community a standard framework for evaluating advancements in engagement modelling.

**Strengths:**

1.	The proposed method is robust, producing a large-scale, high-quality dataset that addresses the gap in image engagement and utility analysis.
2.	The paper is clearly written and well-structured.
3.	The authors evaluate various models, offering valuable insights into the dataset’s application and benchmark characteristics.

**Weaknesses:**

Please see questions.

**Questions:**

1.	A more precise definition and scope of viewer engagement would be helpful, as the current metrics appear to include clicks, likes, and shares. Additionally, detailed statistics of the dataset should be provided to help the community better understand this dataset.
2.	Given the dataset spans over a decade, how is the temporal information being utilized?
3.	While the introduction of the engagement dataset and benchmarking metrics is a valuable contribution, the proposed methods—conditioning image generation, supervised fine-tuning, and reinforcement learning for diffusion alignment—are relatively straightforward adaptations of existing techniques. A more innovative approach to these tasks could elevate the paper’s impact.
4.	The paper could discuss a broader range of use cases and scenarios, such as recommender systems or advertising, beyond standard text-to-image applications, especially given the marketing focus of this work.
5.	An in-depth discussion on potential use cases would be beneficial.
6.	The paper does not clarify whether any collected data contain identifiable information. If they do, obtaining informed consent from individuals would be essential.

**Details Of Ethics Concerns:**

The paper does not clarify whether any collected data contain identifiable information. If they do, obtaining informed consent from individuals and getting the human ethics approval would be essential.

---

> ### Author Response · Authors · 2024-11-28
> **Author Response to Reviewer k6Wd (1/4)**
>
> We are thankful to the reviewer for their encouraging and constructive feedback. Below, we address the reviewer’s questions and comments in detail:
>
> ---
> > Definition of viewer engagement
> ---
> In the context of this paper, viewer engagement refers to the measurable interactions and that a viewer exhibits towards an image, on a given digital platform. These interactions reflect the degree to which the image content resonates with the viewer and elicits actionable behavior. Precisely, in the EngagingImageNet dataset, we consider the account-normalised likes received on a social media post as the metric for viewer engagement.
>
> ---
> > Detailed statistics of the dataset
> ---
> We discuss dataset details and statistics in the following sections of the paper:
> - Section 2: It lists the number of tweets extracted, number of media tweets, number of accounts, time period of collection, counts, average engagement and engagement thresholds for bucketisation of tweets into high, medium, low categories.
> - Table 1: It presents a comparison of EngagingImageNet data size and associated metadata with other datasets containing image preferences
> - Appendix D: It discusses the filtering steps used to prepare EngageNet training data, thresholds for filtering, tweet length, time period, number of accounts and dataset size after filtering,
> - Appendix E: Table 6 presents average number of objects contained, average aesthetic score and average CLIP score of high and low engagement images in the EngagingImageNet dataset. Figure 6 shows that high and low engagement images have a similar distribution of aesthetic scores, further providing evidence that existing text-to-image evaluation metrics like aesthetic score may not be aligned with actual image engagement. Figure 7 displays plots showing variation of number of tweets and likes with time for a few companies in the EngagingImageNet, indicating that engagement may follow seasonal patterns for some companies. We have also added Figure 8, which features a sunburst diagram illustrating the distribution of tweet topics within the EngagingImageNet dataset, showcasing its wide variety of diverse topics.
>
> If you believe any additional analyses could strengthen the paper, we would be happy to incorporate the same into the next draft.
>
> ---
> > how is the temporal information being utilized?
> ---
> - We provide temporal information (posting date) as input to the EngageNet model as our experiments indicate that temporal information also plays a key role in the engagement of an image.
> - Impact of temporal information on Pearson Correlation: In Section 3.2 (lines 270-275), we show results of the ablation study to assess the impact of temporal information. If posting date is not provided as input to the EngageNet model, the correlation of EngageNet’s predictions with actual likes drops (0.62 to 0.53).
> - Seasonal Patterns of Engagement: Figure 7 displays plots showing variation of number of tweets and likes with time for a few companies in the EngagingImageNet dataset, which reveal seasonal patterns in social media engagement, further corroborating the significance of temporal information.
>
> ---
> > The paper does not clarify whether any collected data contain identifiable information.
> ---
> We have discussed PII in Appendix H. We will highlight this better. Concretely, we have implemented measures to remove personally identifiable information (PII). Our dataset is compiled without collecting any personal data, focusing solely on public, non-individualized information. All references to specific users or personal identifiers have been removed. Additionally, we collect only aggregate metrics (e.g., overall user interaction data) to measure engagement trends without compromising individual privacy. Further, we plan to open-source EngagingImageNet in phases starting with smaller releases, engaging with the community continuously such that any issues if present are brought to notice and resolved faster.

---

> ### Author Response · Authors · 2024-11-28
> **Author Response to Reviewer k6Wd (2/4)**
>
> ---
> > discuss a broader range of use cases and scenarios, in-depth discussion on potential use cases
> ---
>
> One of the key contributions of this paper is the training of EngageNet, an engagement-aware vision-language model (VLM) model for predicting viewer engagement over images. This provides a pathway for empowering a gamut of applications. We will add future work and applications in the next paper draft.
>
> In the paper, we demonstrate two applications of the EngageNet model – (1) proposing Engagement Arena, an automated arena to benchmark the engagement of text-to-image models using EngageNet's predicted engagement scores, and (2) Leveraging EngageNet-based reward scores to optimize text-to-image generation for engagement.
>
> Here, we also discuss broader applications of this work beyond the ones already demonstrated in the paper:
> - **Personalization, Campaign Optimization, and Recommender Systems**: EngageNet model can aid marketers and advertisers by selecting/prescribing images among a pool of candidate assets that maximize viewer engagement, making it a vital tool for campaign optimization. EngageNet can be integrated with multimodal recommendation systems to improve the relevance and appeal of suggested images shown to users.
>
> - **Health and Public Awareness Campaigns**: Generating engaging and compelling images for public awareness campaigns, such as health education, can amplify their effectiveness. There are many research works in the public health and medicine literature showing the positive outcomes of health interventions like engaging advertisements [1, 2].
>
> - **Infinite Personalization**: Today, an ad delivery engine has to choose amongst a fixed set of candidates while delivering an ad to an incoming visitor. Thanks to innovations in generative AI models like LLMs and Diffusion models, we can imagine generating many variants of an advertisement. For each incoming visitor, we can use Engagenet to score the variants, thus enabling infinite personalization, where every individual can receive bespoke content, maximizing relevance and engagement.
>
> - **Efficient A/B Testing**: A/B testing is a cornerstone of optimizing marketing campaigns. However, due to the complexity and resource-intensive nature of A/B tests, few marketers are actually able to use them. Among those who do engage in A/B testing, the data-intensive requirements necessary to reach statistical significance often limit them to testing only a few variants at a time. Thus, marketers can employ EngageNet to shortlist and prioritize images with high engagement potential, leading to more efficient and data-driven A/B testing.
>
>
>
> **Future Directions**
>
> Building on the foundational work of this paper, future research could explore the following areas:
> - **Cross-platform engagement optimization**: Extending the framework to incorporate diverse engagement metrics across platforms such as Instagram, YouTube, etc., capturing a broader spectrum of user preferences.
> - **Explaining content factors leading to higher engagement**: Understanding why certain visuals generate higher engagement is crucial for both creators and researchers. Future work could involve enhancing EngageNet and Engagement Arena to capture subtle engagement dimensions such as emotional resonance, visual complexity, and long-term memorability.
> - **Prescribing Content Strategies to Creators**: Beyond explaining engagement factors, future iterations of EngageNet could actively prescribe actionable strategies to creators. For instance, it could suggest specific attributes (e.g., color schemes, composition, or visual themes) that are likely to increase engagement for a particular audience or context, thus empowering creators to produce high-performing content efficiently.
> - **Video Engagement Optimization**: Extending the proposed framework to handle video content could provide insights into optimizing video advertisements, storytelling, and social media clips for engagement. It involves considering dynamic elements like pacing, transitions, and viewer retention metrics, which play a critical role in video content performance.
> - **Fashion industry**: The fashion industry could benefit from integrating engagement-aware content generation into virtual try-on systems. Models could generate visuals tailored to consumer preferences, encouraging them to explore and interact with virtual products. This could also enhance online shopping experiences by presenting customers with styles predicted to resonate with their personal tastes.
> - **Enhanced Recommender Systems**: Engagement-aware images have the potential to enhance recommendation models significantly. By integrating these visuals into recommender systems across domains like retail, food delivery, and streaming services, businesses can present more appealing and relevant suggestions to users, thereby boosting engagement and conversions.
>
> References (continued...)

---

> > ### Author Response · Authors · 2024-11-28
> > **Author Response to Reviewer k6Wd (3/4)**
> >
> > References:
> > - [1] Wakefield, M. A., Loken, B., & Hornik, R. C. (2010). Use of mass media campaigns to change health behaviour. Lancet (London, England), 376(9748), 1261–1271. https://doi.org/10.1016/S0140-6736(10)60809-4
> > - [2] Kite, J., Chan, L., MacKay, K., Corbett, L., Reyes-Marcelino, G., Nguyen, B., ... & Freeman, B. (2023). A model of social media effects in public health communication campaigns: systematic review. Journal of Medical Internet Research, 25, e46345.

---

> > > ### Author Response · Authors · 2024-11-28
> > > **Author Response to Reviewer k6Wd (4/4)**
> > >
> > > ---
> > > > While the introduction of the engagement dataset and benchmarking metrics is a valuable contribution, the proposed methods—conditioning image generation, supervised fine-tuning, and reinforcement learning for diffusion alignment—are relatively straightforward adaptations of existing techniques. A more innovative approach to these tasks could elevate the paper’s impact.
> > > ---
> > > The primary focus of this work is on establishing a robust foundational framework for evaluating image engagement and benchmarking text-to-image models in their ability to generate engaging images. We curate EngagingImageNet, a large-scale, high-quality dataset consisting of user engagement over images. The dataset is instrumental in our study of image engagement in real-world marketing scenarios. We train an engagement-aware vision language model (VLM), called EngageNet, to predict user engagement over images. Using EngageNet’s predicted engagement scores as a reward, we introduce Engagement Arena to benchmark the engagement of text-to-image models.
> > >
> > > We also make initial strides in engagement-driven image generation, providing a pathway to further advance text-to-image modelling. To this end, we explore three methods -- conditioning of text-to-image generation on prompts corresponding to high user engagement, supervised fine-tuning of stable diffusion on high-engagement images, and reinforcement learning to align stable diffusion with EngageNet-based rewards, all of which lead to the generation of more engaging images to varying degrees.
> > >
> > > As this is a relatively nascent area, we aimed to demonstrate introducing the goal of engagement in the text-to-image generation, based on proven methodologies to achieve this. These techniques were selected not only for their effectiveness but also for their accessibility to a broader research community, ensuring that our work could serve as a baseline for future advancements in engagement-driven image generation. Building on our current work, future research works can look into innovating on how to further improve the engagement potential of generated images. We envisage this would be similar to how multiple research works built on top of initial papers to improve image generation quality for metrics like Aesthetic Score [1, 2], PickScore [2,3] and more.
> > >
> > > References:
> > > [1] Black, K., Janner, M., Du, Y., Kostrikov, I., & Levine, S. Training Diffusion Models with Reinforcement Learning. In The Twelfth International Conference on Learning Representations.
> > > [2] Clark, K., Vicol, P., Swersky, K., & Fleet, D. J. Directly Fine-Tuning Diffusion Models on Differentiable Rewards. In The Twelfth International Conference on Learning Representations.
> > > [3] Wallace, B., Dang, M., Rafailov, R., Zhou, L., Lou, A., Purushwalkam, S., ... & Naik, N. (2024). Diffusion model alignment using direct preference optimization. In Proceedings of the IEEE/CVF Conference on Computer Vision and Pattern Recognition (pp. 8228-8238).

---

### Official Review · Reviewer_mpVe · 2024-11-04

**Soundness:** 2
**Presentation:** 2
**Contribution:** 2
**Rating:** 5
**Confidence:** 3

**Summary:**

This paper investigates how current text-to-image generation models fall short of predicting viewer engagement. To address this gap, the authors propose several contributions including (1) EngagingImageNet: A dataset of high-quality enterprise images alongside engagement metrics like user likes, which serve as real-world indicators of viewer engagement, (2) EngageNet: A Vision Language Model (VLM) trained on EngagingImageNet to predict engagement by analyzing contextual factors around each image, such as tweet content, enterprise details, and posting time, and (3) Engagement Arena: A benchmarking platform where EngageNet scores various image generation models based on their capacity to generate engaging images. The paper explores approaches to enhance engagement in generated images, including prompt-based conditioning, fine-tuning, and reinforcement learning with EngageNet-generated reward signals.

**Strengths:**

- The paper presents an interesting application to optimize image generation for viewer engagement, a metric commonly aligned with recommendation and commercial objectives.
- The dataset curation and model development appear thorough. EngagingImageNet is large, high-quality, and potentially valuable for further research on engagement in visual content. The paper includes rigorous validation of EngageNet’s correlation with actual engagement metrics, offering a credible alternative to existing models.

**Weaknesses:**

- As EngagingImageNet is sourced from enterprise accounts on Twitter, the dataset may have an inherent bias toward corporate content. For a model aiming to set a standard for engagement-optimized image generation, more diverse data sources would provide a broader foundation and improve applicability across various domains.
- The effectiveness of engagement-enhancing techniques is primarily measured using EngageNet scores and correlations. Incorporating additional real-world engagement experiments, for example, a small-scale human evaluation study, would strengthen the claims regarding the improvements in viewer engagement and provide additional validation.
- The negative sampling method for training EngageNet is vaguely described. The paper mentions that negative samples are randomly generated by pairing tweets with unrelated images, yet does not explain how the random sampling affects EngageNet's robustness or alignment with true engagement.

**Questions:**

-  Can you provide further detail on how negative samples were generated and their effects on training EngageNet? How does the negative sampling impact model robustness and alignment with actual user engagement?
- Can EngageNet be adapted to predict other engagement metrics (e.g., CTR, shares) in addition to likes?

---

> ### Author Response · Authors · 2024-11-27
> **Author Response to Reviewer mpVe (1/2)**
>
> We sincerely thank the reviewer for their thoughtful and encouraging feedback. Below, we address the reviewer’s questions and comments in detail:
>
> ---
> > bias toward corporate content, more diverse data sources would provide a broader foundation
> ---
> > Can EngageNet be adapted to predict other engagement metrics ?
> ---
> As suggested by the reviewer, we perform experiments to check EngageNet’s generalization to other social media platforms, engagement metrics and data sources. Specifically, we use the FlickrUser dataset [1] which contains ~500K images from ~2.5K users of the popular photo-sharing platform Flickr. For each image, it contains engagement metrics such as “number of favorites”, “number of views”, etc.
>
> Reasons for picking this dataset:
> - To check engagement prediction on non-corporate content.
> - To check generalization capability across others metrics and social media platforms
>
> Out of a total dataset of 500K images, we use only 15K images for finetuning and we use a test set of size 5K.
> - Firstly, as a baseline, we finetune the pretrained LLaVA-1.5 Vision-Language Model (VLM) to predict the “normalised favorites” (i.e.  “number of favorites” / “number of views”) as the engagement score of an image, given contextual information about the image (such as upload date, description, tags, etc.). This model achieves a correlation of only **0.23** with the actual “normalised favorites” of the test set images.
> - Next, we finetune our EngageNet model, an engagement-aware VLM that inherently understands image aspects that contribute to engagement, for the same task as above. We find that our finetuned EngageNet model achieves a strong correlation of **0.53** with the actual “normalised favorites” of images, despite giving minimal training data.
> - This stark performance gap between the two models can be attributed to EngageNet’s prior learnings about image engagement, which enables it to exhibit strong generalization capability across different engagement metrics, social media platforms and even non-corporate content.
>
> References:
> [1] Abdullahu, F., & Grabner, H. (2025). Commonly Interesting Images. In European Conference on Computer Vision (pp. 180-198). Springer, Cham.

---

> > ### Author Response · Authors · 2024-11-27
> > **Author Response to Reviewer mpVe (2/2)**
> >
> > ---
> > > Negative Samples for training EngageNet model
> > ---
> > Negative samples were introduced to enhance EngageNet's ability to discriminate between relevant and irrelevant image-text pairs, improving its overall robustness. The method for constructing such misaligned pairs is as described below.
> >
> > We augment EngagingImageNet with synthetic data samples. For this, we randomly sample 25% tweets from the high and medium likes buckets of each company and pair the tweet with an unrelated image from a different tweet. The corresponding normalized likes is set to a low value, randomly sampled from the range 5-15. The resulting samples are called negative samples. This process ensured that the negative samples presented a challenging scenario for EngageNet, where the image and textual context were semantically misaligned. Thus, adding negative samples in training EngageNet is conceptually analogous to adding negative samples in contrastive learning.
> >
> > **Impact of Negative Samples**: This helps the EngageNet model with the improved sensitivity to contextual alignment of images. Training on negative samples compelled EngageNet to pay closer attention to the semantic relationship between an image and its contextual tweet metadata. EngageNet learns to penalise an image if it is irrelevant or poorly aligned to the tweet context. This ability is crucial to utilize EngageNet as an oracle for evaluation of text-to-image models in the Engagement Arena, such that aesthetically pleasing but irrelevant generated images do not score high on engagement.
> >
> > ---
> > > small-scale human evaluation study
> > ---
> > We have initiated the effort to conduct a user study. We have started the process of Ethics Review Board approval. We aim to incorporate the results of the user study by the camera-ready version deadline.

---

### Official Review · Reviewer_wnMa · 2024-11-04

**Soundness:** 4
**Presentation:** 3
**Contribution:** 4
**Rating:** 8
**Confidence:** 3

**Summary:**

This paper introduces EngageNet, a text-to-image generation model optimization method to improve the audience engagement of the generated images, rather than merely the aesthetic quality and realism. Based on this, the first automated arena, i.e., EngageNet Arena, is proposed to benchmark the engagement of text-to-image models.

**Strengths:**

- The idea of the engagement-optimized image generation is novel, as images are often created to drive user engagement beyond just aesthetic appeal.
- The paper introduces a large dataset of 168 million tweets with images and associated user engagement metrics like likes, paving the way for such research direction.
- Multiple methods are explored to enhance the engagement of text-to-image models, including prompt conditioning, supervised fine-tuning, and reinforcement learning.

**Weaknesses:**

Null

**Questions:**

What are the further direction in this research topic? Given the situation that many labs may lack computational resources, how to make it possible for these researchers to follow your work? Will the EngageNet also be appliable to other research domains, such as multimodal item recommendation?

---

> ### Author Response · Authors · 2024-11-27
> **Author Response to Reviewer wnMa (1/3)**
>
> We sincerely thank the reviewer for their encouraging review and feedback. We try to address the reviewer's questions below:
>
> ---
> > Given the situation that many labs may lack computational resources, how to make it possible for these researchers to follow your work?
> ---
>
> Thank you for raising this important point regarding accessibility for researchers with limited computational resources. We recognize the importance of ensuring that our work can be built upon by a broad community, including labs and student researchers with constrained computational resources.
>
> - While the initial dataset collection and benchmarking phases required substantial computational resources, the subsequent steps, including engagement-driven retrieval, finetuning, and DDPO (for engagement-aligned image generation), can be conducted on relatively modest infrastructure. Therefore, by open-sourcing EngagementArena, EngagingImageNet, and EngageNet, we encourage the research community (especially student researchers) to build upon the released resources and submit their innovations in better engagement-driven retrieval, finetuning and engagement-aligned image generation mechanisms.
>
> - We aim to make EngagementArena as an open source platforms like LMSys and labinthewild.org where students and researchers can contribute and run experiments on public infra.
>   - LMSys is supported by public donations (https://lmsys.org/donations/) from organizations like Huggingface, MBZUAI, Kaggle, NVIDIA, etc. Similarly, a few organizations have privately reached out to kindly lend their resources to make a similar open-source infrastructure. We plan to release this on acceptance.
>
> - Metrics more aligned with engagement than the proposed one: Just as the image aesthetic score prediction landscape evolved from initial papers to more refined and accurate models, researchers can use the EngagingImageNet dataset and build upon the EngageNet model to develop better metrics for engagement. This iterative improvement process will refine the understanding of what drives viewer engagement and lead to more aligned and nuanced models over time.

---

> ### Author Response · Authors · 2024-11-27
> **Author Response to Reviewer wnMa  (2/3)**
>
> > What are the further directions in this research topic? Will the EngageNet also be appliable to other research domains, such as multimodal item recommendation?
>
> One of the key contributions of this paper is the training of EngageNet, an engagement-aware vision-language model (VLM) model for predicting viewer engagement over images. This provides a pathway for empowering a gamut of applications. We will add future work and applications in the next paper draft.
>
> In the paper, we demonstrate two applications of the EngageNet model – (1) proposing Engagement Arena, an automated arena to benchmark the engagement of text-to-image models using EngageNet's predicted engagement scores, and (2) Leveraging EngageNet-based reward scores to optimize text-to-image generation for engagement.
>
> Here, we also discuss broader applications of this work beyond the ones already demonstrated in the paper:
> - **Personalization, Campaign Optimization, and Recommender Systems**: EngageNet model can aid marketers and advertisers by selecting/prescribing images among a pool of candidate assets that maximize viewer engagement, making it a vital tool for campaign optimization. EngageNet can be integrated with multimodal recommendation systems to improve the relevance and appeal of suggested images shown to users.
>
> - **Health and Public Awareness Campaigns**: Generating engaging and compelling images for public awareness campaigns, such as health education, can amplify their effectiveness. There are many research works in the public health and medicine literature showing the positive outcomes of health interventions like engaging advertisements [1, 2].
>
> - **Infinite Personalization**: Today, an ad delivery engine has to choose amongst a fixed set of candidates while delivering an ad to an incoming visitor. Thanks to innovations in generative AI models like LLMs and Diffusion models, we can imagine generating many variants of an advertisement. For each incoming visitor, we can use Engagenet to score the variants, thus enabling infinite personalization, where every individual can receive bespoke content, maximizing relevance and engagement.
>
> - **Efficient A/B Testing**: A/B testing is a cornerstone of optimizing marketing campaigns. However, due to the complexity and resource-intensive nature of A/B tests, few marketers are actually able to use them. Among those who do engage in A/B testing, the data-intensive requirements necessary to reach statistical significance often limit them to testing only a few variants at a time. Thus, marketers can employ EngageNet to shortlist and prioritize images with high engagement potential, leading to more efficient and data-driven A/B testing.
>
>
>
> **Future Directions**
>
> Building on the foundational work of this paper, future research could explore the following areas:
> - **Cross-platform engagement optimization**: Extending the framework to incorporate diverse engagement metrics across platforms such as Instagram, YouTube, etc., capturing a broader spectrum of user preferences.
> - **Explaining content factors leading to higher engagement**: Understanding why certain visuals generate higher engagement is crucial for both creators and researchers. Future work could involve enhancing EngageNet and Engagement Arena to capture subtle engagement dimensions such as emotional resonance, visual complexity, and long-term memorability.
> - **Prescribing Content Strategies to Creators**: Beyond explaining engagement factors, future iterations of EngageNet could actively prescribe actionable strategies to creators. For instance, it could suggest specific attributes (e.g., color schemes, composition, or visual themes) that are likely to increase engagement for a particular audience or context, thus empowering creators to produce high-performing content efficiently.
> - **Video Engagement Optimization**: Extending the proposed framework to handle video content could provide insights into optimizing video advertisements, storytelling, and social media clips for engagement. It involves considering dynamic elements like pacing, transitions, and viewer retention metrics, which play a critical role in video content performance.
> - **Fashion industry**: The fashion industry could benefit from integrating engagement-aware content generation into virtual try-on systems. Models could generate visuals tailored to consumer preferences, encouraging them to explore and interact with virtual products. This could also enhance online shopping experiences by presenting customers with styles predicted to resonate with their personal tastes.
> - **Enhanced Recommender Systems**: Engagement-aware images have the potential to enhance recommendation models significantly. By integrating these visuals into recommender systems across domains like retail, food delivery, and streaming services, businesses can present more appealing and relevant suggestions to users, thereby boosting engagement and conversions.

---

> ### Author Response · Authors · 2024-11-27
> **Author Response to Reviewer wnMa (3/3)**
>
> References:
> - [1] Wakefield, M. A., Loken, B., & Hornik, R. C. (2010). Use of mass media campaigns to change health behaviour. Lancet (London, England), 376(9748), 1261–1271. https://doi.org/10.1016/S0140-6736(10)60809-4
> - [2] Kite, J., Chan, L., MacKay, K., Corbett, L., Reyes-Marcelino, G., Nguyen, B., ... & Freeman, B. (2023). A model of social media effects in public health communication campaigns: systematic review. Journal of Medical Internet Research, 25, e46345.

---

### Meta-Review · Area_Chair_9Xm2 · 2024-12-20

**Metareview:**

This paper propose several contributions including (1) EngagingImageNet: a large scale dataset of tweet images alongside engagement metrics like user likes, (2) EngageNet: A VLM trained on EngagingImageNet to predict engagement and (3) Engagement Arena: A benchmarking platform to use EngageNet scores to evaluate multiple generation models on their capacity to generate engaging images.

The major strengths of the work are: 1)  the idea and direction of engagement-optimized image generation is quite important and novel 2) The large scale data set EngagingImageNet and the EngageNet (assume it will be released) is an important contribution and should be helpful to the research community 3) Multiple strategies for enhancing the engagement, during run-time and train-time, are discussed, and shown to have notable improvement.

The major weaknesses are: 1) data set bias since EngagingImageNet contains mostly of corporate images 2) lack of human evaluation study to show the engagement is indeed improved 3) some technical parts like negative sampling are not clear enough. 4) some related works are mising as pointed out reviewer wzeb.

Even though the data set EngagingImageNet may have some bias, but given its novelty and scale, I feel it is quite helpful to the research community. Moreover, multiple methods are explored to improve the engagement of the generated images. I agree a human evaluation study will make the experiment results more convincing, but current results also provide enough interesting observations and insights.

**Additional Comments On Reviewer Discussion:**

Reviewer wnMa asked questions about making it possible for researchers with limited resources to follow the work, and the authors addressed it quite well by what these researchers can benefit from open-sourced EngagementArena, EngagingImageNet, and EngageNet. Authors also provided good answers of future directions etc.

Reviewer mpVe asked about data set bias and whether the proposed method can generalize to other engagement data? Authors provided experiments on FlickrUser dataset to show the generazation.

Reviewer k6Wd asked multiple questions about technical and experiment details, and reviewer wzeb asked questions about related works and some additional experiments, most of which are addressed well by the authors.

---

### Decision · Program_Chairs · 2025-01-22

Accept (Poster)